# Measurement report: Shipborne observations of black carbon aerosols in the western Arctic Ocean during summer and autumn 2016–2020: impact of boreal fires

Yange Deng[1], Hiroshi Tanimoto[1], Kohei Ikeda[1], Sohiko Kameyama[2], Sachiko Okamoto[1,a], Jinyoung Jung[3], Young Jun Yoon[3], Eun Jin Yang[3] and Sung-Ho Kang[3]

1 National Institute for Environmental Studies, Tsukuba, Japan

2 Hokkaido University, Sapporo, Japan

Korea Polar Research Institute, Incheon, Korea

a now at: University Paris Est Creteil and Université de Paris Cité, CNRS, LISA, 94010 Créteil, France

*Correspondence to*: Hiroshi Tanimoto (tanimoto@nies.go.jp)

Black carbon (BC) aerosol is considered one of the important contributors to the fast climate warming and snow and sea ice melting in the Arctic. Yet the observations of BC aerosols in the Arctic Ocean have been limited due to infrastructural and logistical difficulties. We observed BC mass concentrations ($m_{BC}$) using light absorption methods on board the icebreaker R/V *Araon* in the Arctic Ocean (166° E–156° W and <80° N) as well as the North Pacific Ocean in summer and early Autumn of 2016 to 2020. The levels, interannual variations and pollution episodes of $m_{BC}$ in the Arctic were examined, and the emission sources responsible for the high BC episodes were analyzed with global chemistry-transport model simulations. The average $m_{BC}$ in the surface air over the Arctic Ocean (72–80° N) observed by the 2019 cruise exceeded 70 ng m$^{-3}$, which was substantially higher than cruises in other years (approximately 10 ng m$^{-3}$). The much higher $m_{BC}$ observed in 2019 was perhaps due to more frequent wildfires occurring in the Arctic region than in other years. The model suggested that biomass burning contributed most to the observed BC by mass in the western Arctic Ocean and the marginal seas. For these five years, we identified 10 high BC episodes north of 65° N, including one in 2018 that was associated with co-enhancements of CO and $CH_4$ but not $CO_2$ and $O_3$. The model analysis indicated that certain episodes were attributed to BC containing airmasses transported from boreal fire regions to the Arctic Ocean, with some transport occurring near-surface and others in the mid-troposphere. This study provides crucial datasets on BC mass concentrations and the mixing ratios of $O_3$, $CH_4$, CO, and $CO_2$ in the western Arctic Ocean regions and highlights the significant impact of boreal fires on the observed Arctic BC during the summer and early autumn months.

# 1  Introduction

The annual average surface temperature increase in the Arctic is more than three times the global average increase, resulting in a rapid decline of Arctic sea ice extent in all months, a decrease in extreme cold events, and other ecosystem changes (AMAP, 2021a; IPCC, 2021). While global anthropogenic carbon dioxide ($CO_2$) emissions play the dominant role in driving Arctic climate change, short-lived climate forcers (SLCFs) – such as methane ($CH_4$), ozone ($O_3$), nitrogen oxides, and aerosols – have considerable potential to mitigate climate warming in the Arctic (AMAP, 2015, 2021b). Arctic aerosol chemical composition may include black carbon (BC), sulfate ($SO_4$), nitrate ($NO_3$), organics, sea-salt, and mineral dust (Sakerin et al., 2015; AMAP, 2021b; Schmale et al., 2022). Particularly, BC aerosols in the Arctic atmosphere can absorb solar radiation directly which causes direct and/or semi-direct climate forcing (AMAP, 2011). Besides, BC aerosols can also act as cloud condensation nuclei (CCN) which causes indirect climate forcing (AMAP, 2011; McFarquhar et al., 2011). When deposited onto snow/ice surface, BC can also affect the radiation budget due to reduction of the surface albedo, leading to an acceleration in the melting of snow and ice (AMAP, 2011). According to Oshima et al. (2020), BC in the Arctic provides the second largest contribution to the positive effective radiative forcings after $CO_2$. Therefore, BC plays an important role in Arctic climate forcing.

Systematic monitoring of BC in the Arctic is critical to provide a better scientific basis for making mitigation policies. Long-term BC observations have been carried out at ground-based Arctic observatories on continental Arctic, such as Utqiaġvik, Alert, Zeppelin, Summit, Pallas and Tiksi, and Gruvebadet (e.g., Stohl et al., 2013; Schmale et al., 2022).  Whereas these long-term datasets provided essential information on the seasonal and interannual variations of BC in the Arctic (e.g., Schmale et al., 2022), they are limited in representing the spatial variation of BC in the Arctic Ocean. Such limitations can be partially compensated for by shipborne and airborne observations. Airborne observations have illustrated the vertical distributions of BC above the Arctic Ocean surface (e.g., Schulz et al., 2019; Ohata et al., 2021a; Jurányi et al., 2023). Meanwhile, shipborne observations have facilitated in situ measurements in the remote Arctic Ocean, especially in summer and autumn when the Arctic sea ice is at the minimum, making access to the Arctic Ocean easier (Xie et al., 2007; Sierau et al., 2014; Kim et al., 2015; Sakerin et al., 2015, 2021; Taketani et al., 2016, 2022; Popovicheva et al., 2017; Ding et al., 2018; Terpugova et al., 2018; Shevchenko et al., 2019; Pankratova et al., 2020; Park et al., 2020; Nagovitsyna et al., 2023). In addition, Boyer et al. (2023) measured the BC mass concentration in the central Arctic (>80° N) for a whole year from September 2019 to October 2020. These shipborne studies have provided BC mass concentration results used for model evaluation in the Arctic Ocean (e.g., Whaley et al., 2022). They also revealed important characteristics of the spatial distribution of BC in the Arctic Ocean, demonstrating that BC concentration diminishes in the

northern direction and decreases as distance from the continent increases (Xie et al., 2007; Sakerin et al., 2015, 2021). The year-round observation in the central Arctic by Boyer et al. (2023) indicated that seasonal changes in BC are similar to those of the Arctic continent, but the changes are larger, with high values in winter and spring – the Arctic Haze season (Barrie, 1986), and low values in summer and early autumn.

However, most of these studies were limited to the North Atlantic and Eurasian Arctic Seas (Sierau et al., 2014; Sakerin et al., 2015, 2021; Popovicheva et al., 2017; Terpugova et al., 2018; Shevchenko et al., 2019; Pankratova et al., 2020; Nagovitsyna et al., 2023; Boyer et al., 2023) and the Bering, Chukchi, and Beaufort Seas (Xie et al., 2007; Kim et al., 2015; Sakerin et al., 2015; Taketani et al., 2016, 2022; Ding et al., 2018; Nagovitsyna et al., 2023). To our knowledge, Xie et al. (2007) and Ding et al. (2018) are the only two studies that reported BC observations in the western Arctic Ocean north of 74° N, and Shevchenko et al. (2019) is the only study related to BC observation in the East Siberian Sea. Furthermore, BC in Xie et al. (2007) was only qualitatively quantified. Therefore, for a better understanding of the spatial-temporal variations of BC in the Arctic Ocean and better model constraint, continuous shipborne observations of BC in the Arctic marine boundary layer especially in the western central Arctic Ocean and East Siberian Sea, where data coverage is sparse, are highly necessary under the rapidly changing Arctic environments (AMAP, 2021a; Whaley et al., 2022; Jurányi et al., 2023).

The accurate location of BC sources is another important step toward mitigation measures. Atmospheric modelling is indispensable in understanding the distributions and sources of BC in the Arctic quantitatively. Current atmospheric models still have difficulties in accurately reproducing the BC abundance in the Arctic (e.g., Whaley et al., 2022; Jurányi et al., 2023). The main obstacles include poor understanding of long-range transport, vertical mixing, deposition, and emissions (e.g., Ikeda et al., 2017; Whaley et al., 2022). Preexisting modeling studies combined with field observations indicate that biomass burning from Siberia as well as Alaska and Canada contributed the most to surface BC mass concentration during summer and early autumn (e.g., Zhu et al., 2020; Popovicheva et al., 2022). In addition, according to McCarty et al. (2021), wildfire emissions of BC above 60° N have increased from 2010 to 2020 and open biomass burning contributed 56 % of BC emissions above 65° N in 2020. In the context of climate change, the likelihood of extreme fire weather in the Arctic will increase (McCarty et al., 2021). Consequently, the impact of BC emissions from boreal vegetation fires on the Arctic atmospheric BC may increase (AMAP, 2021a, b). Therefore, continual studies combining field observations and modelling simulations on the impact and transport of biomass burning BC in boreal areas to the Arctic Ocean are urgently needed.

In this study, to enhance comprehension of the distribution and sources of BC in the Arctic, the mass concentration of BC ($m_{BC}$) was monitored across five round-trip expeditions conducted between the North Pacific Ocean and the Arctic Ocean during the summer and early autumn of 2016–2020. Based on the

observations, the spatial-temporal variations of $m_{BC}$ were characterized and the background $m_{BC}$ in the western Arctic Ocean was estimated. The observations were compared with BC tagged-tracer simulations using GEOS-Chem (Ikeda et al., 2017). The sources of observed BC and air masses containing high BC mass concentrations were interpreted based on GEOS-Chem model and back trajectory analysis. The results from this study demonstrate the significant impacts of boreal fires on the observed BC in the western Arctic Ocean and its marginal seas.

## 2   Shipborne observations

The shipborne observations were conducted in summer and autumn in the years of 2016 to 2020 (Fig. S1a) on board the icebreaker R/V *Araon* operated by the Korea Polar Research Institute (KOPRI), South Korea. The air intake was set at the handrail of the front upper deck to prevent contamination from ship exhaust pollution. Furthermore, detailed information regarding data filtering techniques to mitigate the impact of ship exhaust will be provided later. A cyclone was attached at the intake to selectively sample $PM_{2.5}$ aerosols. The total air flow rate was 10 L/min.

A continuous soot monitoring system (COSMOS, model 3130, KANOMAX, Japan) and an Aethalometer (model AE22, serial number 1057:1010, Magee Scientific Co., USA) were used during the cruises to measure the mass concentrations of BC aerosols. Whereas both instruments use light absorption methods, COSMOS was equipped with a 400 °C heated inlet line. This feature effectively eliminated interference from volatile non-refractory aerosol chemical species internally mixed with BC, ensuring a high accuracy of $m_{BC}$ measurement. This aspect has been critically assessed in previous studies (Ohata et al., 2019; Sinha et al., 2017). Consequently, COSMOS measurements differ from traditional light absorption methods, where the mass concentration of BC is referred to as equivalent BC (eBC, Petzold et al., 2013). Therefore, instead of using eBC, the term BC can be used for COSMOS data in a general sense (Ohata et al., 2019). Henceforth, when comparing data from the two different instruments, we will use $m_{eBC}$ to represent the BC mass concentration measured with the Aethalometer during the 2017, 2018, and 2020 cruises, and $m_{BC}$ (COSMOS) to represent the BC mass concentration measured with COSMOS during the 2016–2019 cruises. Otherwise, BC mass concentration is denoted as $m_{BC}$ for simplicity.

COSMOS monitors changes in transmittance of 565 nm wavelength LED light across an automatically advancing quartz fiber filter tape. To achieve measurements with high sensitivity and a lower detectable light absorption coefficient, COSMOS uses a double-convex lens and optical bundle pipes to maintain high light intensity and signal data are obtained at 1000 Hz. In addition, its sampling flow rate (0.9 L min$^{-1}$) and optical unit temperature were actively controlled. The measurement interval was set to 1 min, which was then averaged to 1 h for further analysis. The default mass absorption cross section (MAC) of 10 m$^2$ g$^{-1}$

was applied for the derivation of $m_{BC}$. The lowest detection limit of COSMOS at 1 min time resolution is 50 ng m$^{-3}$. On an hourly basis, COSMOS can measure $m_{BC}$ in the range of 1–3000 ng m$^{-3}$ with an average accuracy of ~10 %, as compared with measurements by a single particle soot photometer (SP2) (Moteki and Kondo, 2010); and its sensitivity to the changes in the BC size distributions was less than 10 %, within the typical BC sizes in ambient atmosphere (Ohata et al., 2019). The SP2 was often used as a reference instrument in previous studies (e.g., Ohata et al., 2019; Sinha et al., 2017). Further details about the measurement principles of COSMOS can be found in previous studies (Ohata et al., 2019; Kondo et al., 2009).

The Aethalometer uses the absorption of light at a wavelength of 880 nm by ambient aerosols collected on a quartz filter tape to determine the BC concentration. The flow rate was set to 5 L min$^{-1}$ and the accumulation area of the filter is 1.67 cm$^2$. The filter was set to change every 24 hours to minimize the loading effects. The data integration time was set to 5 min. For further analysis, hourly averages were used to minimize noise levels under clean atmospheric conditions. The default manufacturer-provided MAC value of 16.6 m$^2$ g$^{-1}$ was applied for all analyses since the study area covers a wide range of latitudes. The manufacturer's particle-free zero air testing meets a 24-h mean detection limit of 20 ng m$^{-3}$ and a 5-min standard deviation limit of ±30 ng m$^{-3}$. Comparison between $m_{eBC}$ and $m_{BC}$ (COSMOS) for cruises in 2017 and 2018, when both data are available, shows that the two data are in high consistency (Pearson correlation coefficient $R>0.96$) and that $m_{eBC}$ was 1.3–2.5 times $m_{BC}$ (COSMOS) (Fig. S2). Previous studies also show that the default parameter settings of the Aethalometer, as mentioned above, may cause the obtained BC mass concentrations to be 1–3 times the mass measured by SP2, depending on the sources and mixing states of the BC aerosols (Wang et al., 2014; Sharma et al., 2017; Laing et al., 2020). Due to the above reasons, the AE22 data in this study are mainly used as a reference. Hereinafter, for cruises conducted from 2016 to 2019, the analysis primarily relied on COSMOS data. In the case of the 2020 cruise, when only AE22 data was available, AE22 data was utilized for the analysis.

In addition, the atmospheric mixing ratios of $CH_4$, carbon monoxide (CO), and $CO_2$ were monitored using a cavity ring-down spectrometer (CRDS) - the Picarro G2401 gas concentration analyzer (Picarro, Inc., USA) when the icebreaker R/V *Araon* was in the Arctic Ocean (North of 72° N) during the cruise in 2018. The Picarro G2401 analyzer was calibrated by running the standard $CH_4$ gas (RIGAS, Korea) for 8 min every day. The $CH_4$, CO, and $CO_2$ data during the instrument calibration period were omitted. The $CH_4$, CO, and $CO_2$ data were averaged to 1 min before being further analyzed. The mixing ratios of $O_3$ were determined using ultraviolet absorption spectroscopy during the cruises in 2017 and 2018 with a time resolution of 1 min. The $O_3$ monitor (Model 1100, Dylec Inc., Japan) utilized absorption at 253.7 nm emitted by a low-pressure mercury lamp and was calibrated through intercomparison with a reference photometer,

which was referenced to the Standard Reference Photometer (SRP) #2 at the National Institute of Standards and Technology (NIST). Those gaseous data were used to assist the analysis on BC sources during high BC episodes in 2018 (Sect. 4.4.2). The $O_3$ data were also used to scrutinize the possible contamination from ship emissions as explained in the next paragraph. Statistics of those gaseous data are shown in Appendix Table A1; times series and concentration distributions of those data along the cruise tracks are presented in Figs. 6-8 and S18.

To avoid the influence of ship exhausts, we only used 1- or 5-min data records that occurred when the 1-min wind direction and speed relative to the ship's course were within ±60° of the bow and >3 m s$^{-1}$, respectively, for continuous 10 min centered around the current 1- or 5-min data record. Furthermore, for the 2017 and 2018 cruise, when the atmospheric mixing ratio of $O_3$ was recorded (Fig. S18), the 1-min COSMOS BC and $O_3$ data were further scrutinized for the possible contamination of ship exhausts considering the $O_3$ titration effect by NO from ship emissions (Pfannerstill et al., 2019). When $O_3$ decreased and BC increased at the same time, both 1-min BC and $O_3$ data were considered invalid. Accordingly, 41–57 % of the observed 1- or 5-min BC data, 56 % of 1-min $O_3$ data, and 63 % of 1-min $CH_4$, CO, and $CO_2$ data were removed from the analysis. It is noteworthy that the additional scrutiny based on the $O_3$ criteria had minimal impact on the overall characteristics of the observed BC by COSMOS. This screening process resulted in the exclusion of less than 0.3 % and 0.4 % of the total valid data in 2017 and 2018, respectively. Furthermore, hourly values are only calculated when there are more than 40 minutes of valid data records in an hour, by averaging the 1-min or 5-min values within that hour. Within the hourly BC mass concentration data, 5–13 % of COSMOS data fall below its detection limit.

## 3   Model simulations

Tagged tracer simulations of BC using the global chemistry transport model GEOS-Chem (v13.1.2; Bey et al., 2001; Ikeda et al., 2017) were performed to assist in the interpretation of the sources and transport paths of observed BC in the Arctic Ocean. The horizontal resolution of GEOS-chem was 2° × 2.5° with 47 vertical layers from the surface to 0.01 hPa. The meteorological data was supplied by Modern-Era Retrospective analysis for Research and Applications, Version 2 (MERRA-2). Two BC tracers, namely anthropogenic BC (BCan) and biomass burning BC (BCbb), were defined for the simulations. The Evaluating of the Climate and Air Quality Impacts of Short-Lived Pollutants version 6b (ECLIPSEv6b) was adopted as anthropogenic emission source (Klimont et al., 2017). The Global Fire Emissions Database with small fires (GFED v4.1s) with 0.25° × 0.25° of spatial resolution and daily temporal resolution was applied as biomass burning emission source (van der Werf et al., 2017). In the following section, the simulated total BC mass

concentration is noted as $m_{BC,S}$, and the simulated BC mass concentrations contributed by anthropogenic and biomass burning sources were noted as $m_{BC,SAN}$ and $m_{BC,SBB}$, respectively.

Furthermore, backward trajectories were generated using the NOAA Air Resources Laboratory Hybrid Single-Particle Lagrangian Integrated Trajectory model (HYSPLIT; Stein et al., 2015) to aid in interpreting the sources of the observed BC and identifying background periods in the Western Arctic Ocean. These trajectories were calculated with a 1-hour time step, initiated at the ship positions with starting heights of 10, 500, and 1000 m above model ground level, and extended for 5 days. The selection of a 5-day duration allows for identifying potential source regions of high BC episodes (Sect. 4.4) while ensuring trajectory accuracy (Backman et al., 2021). The meteorological data used for HYSPLIT was the NCEP's GDAS data, featuring a horizontal resolution of 1° × 1° and 24 pressure levels extending from the ground to 20 hPa in the vertical direction. Note that for the source interpretation of the observed BC, only back trajectories starting at 500 m above model ground level were employed. Trajectories starting at 10, 500, and 1000 m above the model ground level were used for background period identification.

## 4 Results and discussion

### 4.1 Spatial and temporal variations of BC mass concentrations

Figure 1 shows the shipborne observation cruise tracks North of 64° N during 2016–2020. Spatial distributions of the observed BC mass concentrations along the cruise tracks of respective years are indicated by filled color circles in Figs. 1b-f. For all the years, the cruises in the Arctic Ocean took place during August and early September, covering the region of 166° E –156° W and ≤80° N (Fig. 1a; Table 1). The cruise region in the Arctic in this study either fully or partially covered the shipborne research regions in previous studies by Taketani et al. (2016, 2022), Xie et al. (2007), Dall'Osto et al. (2020), Park et al. (2020), and Ding et al. (2018).

The temporal-spatial distribution of BC mass concentrations along the whole cruise tracks in respective years can be found in Fig. S1. To further investigate the spatial and temporal variations of observed BC, the $m_{BC}$ in each cruise were categorized into three groups according to the latitude of the observations, i.e., South of 52° N (in the North Pacific Ocean), North of 72° N (mainly in the Canada Basin and the east part of the East Siberian Sea, which are noted as western central Arctic Ocean in the following sections of this study), and between 52 and 72° N (mainly in the Bering, Chukchi, and Beaufort Seas). They were statistically analyzed and the results are presented in Fig. 2. Note that the grouping mentioned here does not comply with the latitudinal constraints (i.e., north of 65° N) used to select high BC episodes in Sect. 4.4. Time series of the $m_{BC}$ and ship latitudes in each cruise are presented in Fig. 3. In general, $m_{BC}$ in high latitude regions were relatively low, consistent with previous studies (Sakerin et al., 2021) demonstrating a

decrease in $m_{BC}$ with increasing latitude. Additionally, high latitude regions showed fewer temporal-spatial variations compared to low latitude regions. However, frequent high $m_{BC}$ spikes were also observed at high latitudes in 2019. The high $m_{BC}$ observed in lower latitude regions from the North Pacific Ocean to the southern Chukchi Sea near the Bering Strait can be explained by the fact that East Asia is the largest BC source region in the world (Ikeda et al., 2022) and that biomass burning in boreal regions including Siberia, Alaska, and Canada is also a large BC source in summer (Zhu et al., 2020).

Significant but not regular interannual variation of $m_{BC}$ was observed in regions South of 52° N. The highest mean and median $m_{BC}$ values were observed in 2018 and 2017, followed by 2019, 2020, and 2016 (Fig. 2a). At regions between 52 and 72° N and North of 72° N, except the year 2019, $m_{BC}$ variations among other years were not evident (Fig. 2a). The median values of $m_{BC}$ at the former region were 10–12 ng m$^{-3}$ except for the cruise in 2019, when it was around 17 ng m$^{-3}$; at the latter region, the median values were 3–4 ng m$^{-3}$ except for 2019, when it was around 15 ng m$^{-3}$. The higher BC concentration and more frequent high BC spikes in 2019 than other years at the Arctic Ocean and marginal sea regions were likely affected by more frequent outflows of smoke from boreal vegetation fires during the cruise observation period (Sakerin et al., 2020). This is supported by a few studies. For example, Antokhina et al. (2023) reported intensive fire activities during 3 July to 12 August 2019 in Eastern Siberia (95–120° E); Bhatt et al. (2021) reported extreme fire activity started in mid-August in Southcentral Alaska due to the extreme conditions of hot summer temperature and prolonged drought; Voronova et al. (2020) reported that the total burned-out areas and the amounts of emissions of fine aerosols in Siberia were abnormally high in 2019 especially in August; Chen et al. (2023) reported that unprecedented vegetation fires were observed in the eastern Siberia and Alaska in 2019; and Hayasaka (2022) reported that the number of hotspots in summer season in the Arctic region in 2019 was much greater than those in 2016–2018 and 2020. In addition, at Utqiaġvik observatory (Fig. 1a), the nearest surface station to the cruise regions in the Arctic Ocean of this study, the interannual variation of BC mass concentrations measured by a similar COSMOS instrument and absorption coefficient at 550 nm measured by two other filter-based absorption photometers in August and September also presented higher values in 2019 than in other years (Figs. S4-5; Ohata et al., 2021b; https://ads.nipr.ac.jp/dataset/A20201120-001; last access: 8 September 2022.), which is consistent with the interannual variations of BC mass concentrations observed in this study.

The BC mass concentration measured in the western central Arctic Ocean is comparable to some of those in previous shipborne observation studies (Table 2), most of which adopted aethalometer methods, except for those conducted by Taketani et al. (2016, 2021). In this study, the median and mean (± 1 standard deviation) $m_{eBC}$ measured with the AE22 in August 2020 were 3.4 and 14 (±35) ng m$^{-3}$, respectively; the values are close to those measured in the central Arctic Ocean during the same period using a AE33

Aethalometer, where the median and mean ($\pm$ 1 standard deviation) values were 6.5 and 10 ($\pm$22) ng m$^{-3}$, respectively (Boyer et al., 2023). In addition, the mean ($\pm$ 1 standard deviation) $m_{BC}$ (COSMOS) in August and early September 2016 was 10 ($\pm$11) ng m$^{-3}$, aligning with the $m_{eBC}$ value of 23 ($\pm$55) ng m$^{-3}$ obtained in late July and August 2016 by Ding et al. (2018), considering the relative uncertainty factor of 1–3 for the Aethalometer as discussed in Sect. 2. However, this $m_{BC}$ (COSMOS) value is 10 times higher than that reported by Taketani et al. (2022). The large difference was likely caused by the spatial and temporal difference between the measurements in the two studies. The cruise routes in this study covered part of the East Siberian Sea region, whereas that in Taketani et al. (2022) was within the Chukchi and Beaufort Sea regions; and the cruise in this study occurred mainly in August whereas that in Taketani et al. (2022) mainly in September. Resultingly, different airmasses containing different BC concentrations could have been observed by this study and Taketani et al. (2022). Therefore, caution on the temporal and spatial ranges should be taken when comparing the mass concentrations of BC observed in the Arctic Ocean. It is noted that the COSMOS can measure the BC mass concentration in the Arctic with ~10 % accuracy compared with SP2 in 1 h time resolution (Ohata et al., 2019), as mentioned in Sect. 2, therefore the instrument difference should not have influenced the comparison between this study and Taketani et al. (2022) largely. Furthermore, the $m_{BC}$ (COSMOS) measured in this study is lower than most of the $m_{eBC}$ observed in the Eurasian Arctic Seas, except for that observed in the Laptev Sea in 2018 (Pankratova et al., 2020). This is also likely caused by differences in air mass resources.

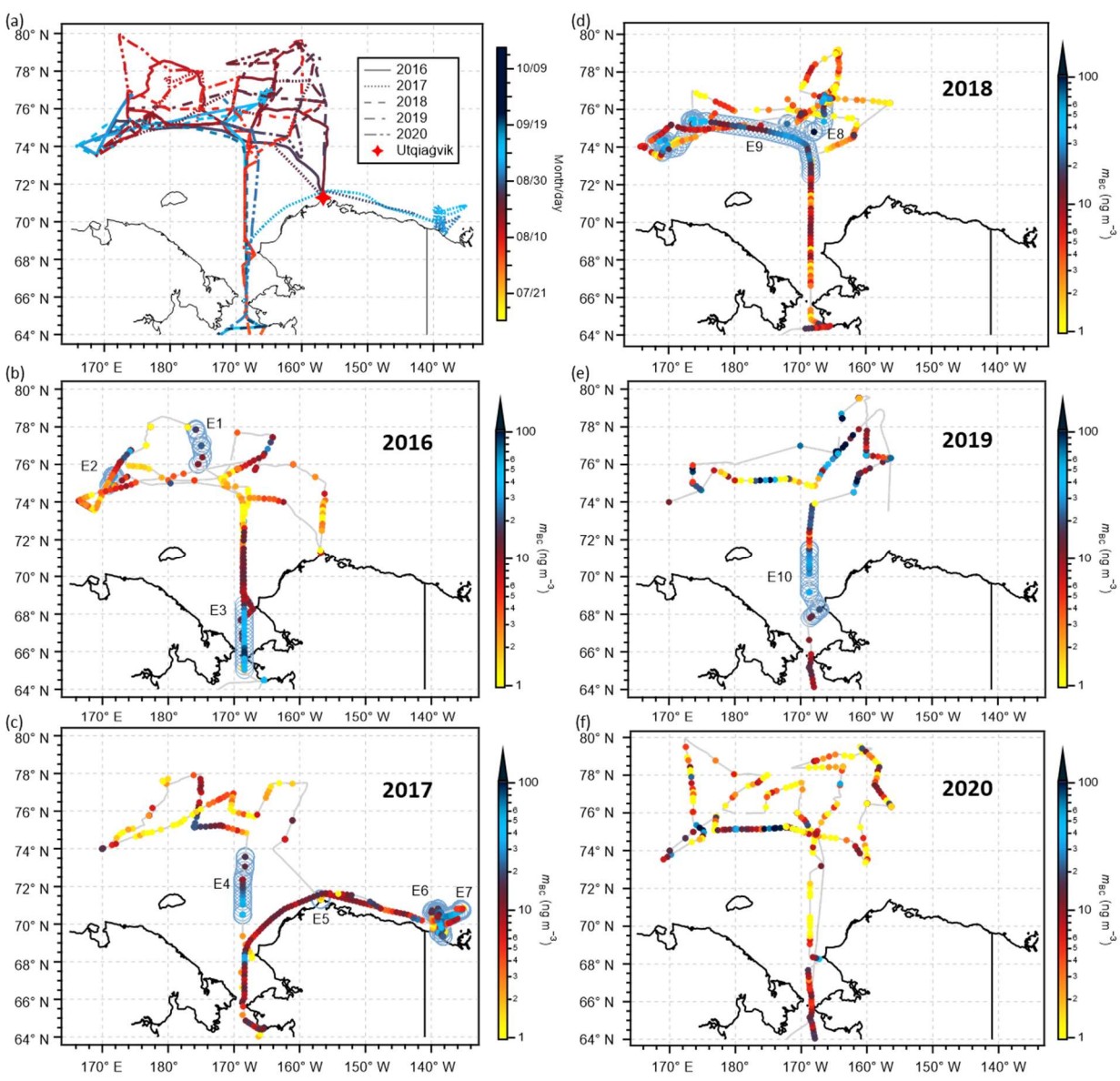

**Figure 1** Shipborne observation cruise tracks North of 64° N during 2016–2020. (a) Color indicates month/day. Star marker indicates the location of the Utqiaġvik observatory (71.29° N, 156.75° W). (b-f) Spatial distribution of BC mass concentrations along the cruise tracks in respective years. The grey line represents the cruise track, and the filled color circle superimposed on the track indicates the BC mass concentration. The $m_{BC}$ presented here is at 1 h time resolution and the data influenced by ship exhaust has been removed. In panels (b-e), ship positions during the 10 episodes (E1-E10) were marked along the ship tracks as open circles. The temporal and spatial distribution of BC mass concentrations along the whole cruise tracks in respective years can be found in Fig. S1.

**Table 1** Time and space coverage of R/V *Araon* and overall and background BC mass concentrations in the Arctic Ocean (≥72° N).

| Year | | 2016 | 2017 | 2018 | 2019 | 2020 |
|---|---|---|---|---|---|---|
| **Period (month/day)** | | 08/08–09/09 | 08/09–08/25 | 08/06–09/18 | 08/08–08/27 | 08/06–08/31 |
| **Latitude (°)** | | +72–+79 | +72–+78 | +72–+79 | +72–+80 | +72–+80 |
| **Longitude (°)** | | +166––156 | +170––159 | +166––156 | +170––156 | +169––156 |
| $m_{BC}$ | **Overall** | 10(±11) | 6.6(±6.7) | 7.8(±15) | 73(±210) | 14(±35)[b] |
| **(ng m$^{-3}$)[a]** | **Background** | 2.8(±2.6) | 9.8(±6.3) | 2.1(±2.5) | 14(±11) | 5.5(±7.0)[b] |

[a] mean(±standard deviation)

[b] $m_{eBC}$.

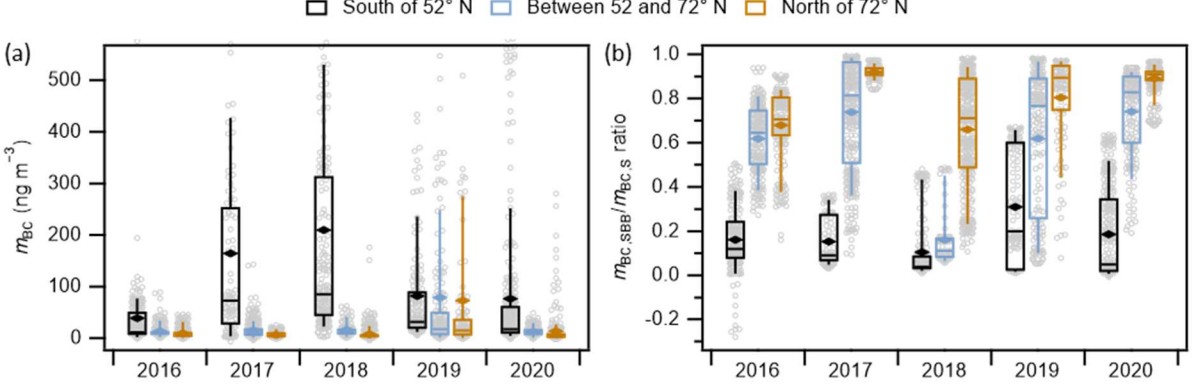

**Figure 2** Box plots of (a) the observed BC mass concentration and (b) the model simulated ratio of biomass burning BC to total BC ($m_{BC,SBB}/m_{BC,S}$) along the ship tracks at latitudes south of 52° N, north of 72° N, and between 52 and 72° N for respective cruises in 2016–2020. Lower whisker – 9th percentile; upper whisker – 91st percentile; box bottom – first quartile; box top – third quartile; line in the box – median value; solid diamond marker – arithmetic mean; open circles – individual data. All data presented here is at 1 h time resolution and the data influenced by ship exhaust has been removed. The full-scale panel (a) and a zoomed-in view of panel (a) with the y-axis maximum set to 80 ng m$^{-3}$ are shown in Fig. S3.

**Table 2** BC mass concentrations based on shipborne observations from previous studies.

| Concentration (ng m$^{-3}$)[a] | Period | Area | Method | R/V | Data Source |
|---|---|---|---|---|---|
| 20±9 | 7-29 Sep, 2013 | Western Arctic Ocean (Utqiaġvik, Beaufort Sea, Nome, and Chukchi Sea) | Aethalometer (AE22: 880 nm) | Araon | Kim et al., 2015 |
| 23±55 | 25 Jul-31 Aug, 2016 | Western Arctic Ocean (≤82.88° N; 180-136° W) | Aethalometer (AE-31: 880 nm) | Xuelong | Ding et al., 2018 |
| 1.0±1.2 | 6-25 Sep, 2014 | Western Arctic Ocean (70-75° N; ~170~156° W) | laser-induced incandescence method (SP2) | Mirai | Taketani et al., 2016, 2022 |
| 0.9±1.4 | 6-30 Sep, 2015 | Western Arctic Ocean (70-~75° N; ~170~153° W) | laser-induced incandescence method (SP2) | Mirai | Taketani et al., 2022 |
| 0.7±1.8 | 3-20 Sep, 2016 | Western Arctic Ocean (70-~74° N; ~170~152° W) | laser-induced incandescence method (SP2) | Mirai | Taketani et al., 2022 |
| 50±20 | 9-24 Aug, 2013 | Chukchi and East Siberian seas (69–71° N) | Aethalometer (MDA-02: 460, 530, 590, and 630 nm) | Professor Khljustin | Sakerin et al., 2015 |
| 140±100 | 18–21 Aug, 2013 | Barents Sea near Kola Peninsula coasts (68–71° N) | Aethalometer (MDA-02: 460, 530, 590, and 630 nm) | Akademik Fedorov | Sakerin et al., 2015 |
| <30 | 9-25 Oct, 2015 | Southeastern Barents Sea | Aethalometer (MSU-CAO: 450, 550, and 650 nm) | Akademik Treshnikov | Popovicheva et al., 2017; Shevchenko et al., 2019 |
| 36±9.2 | 24 Aug, and 29 Sep, 2017 | Southeastern Barents Sea | Quartz fiber filter samples subjected to Aethalometer analysis | Akademik Mstislav Keldysh | Shevchenko et al., 2019 |
| 54 | 19 Aug, 2018 | Southeastern Barents Sea | Quartz fiber filter samples subjected to Aethalometer analysis | Akademik Mstislav Keldysh | Pankratova et al., 2020 |
| 14 | 17-18 Sep, 2018 | Southeastern Barents Sea | Quartz fiber filter samples subjected to Aethalometer analysis | Akademik Mstislav Keldysh | Pankratova et al., 2020 |
| 60±20 | 21–23 Aug and 19–21 Sep, 2013 | Barents Sea (71–81° N) | Aethalometer (MDA-02: 460, 530, 590, and 630 nm) | Akademik Fedorov | Sakerin et al., 2015 |
| 46±13 | Jul 29-Aug 9, 2017 | Barents Sea | Quartz fiber filter samples subjected to Aethalometer analysis | Akademik Mstislav Keldysh | Shevchenko et al., 2019 |
| 37±68 | Summar and Autumn of 2007-2020 | Barents Sea | MDA Aethalometer or quartz fiber filter samples analyzed by absorption photometers | Akademik Mstislav Keldysh, Akademik Fedorov, Akademik Tryoshnikov, Professor Molchanov, or Professor Multanovsky | Sakerin et al., 2021 |
| 20±10 | 24 Aug–18 Sep, 2013 | Arctic Ocean (77–84° N, 80–160° E) | Aethalometer (MDA-02: 460, 530, 590, and 630 nm) | Akademik Fedorov | Sakerin et al., 2015 |

| | | | | | |
|---|---|---|---|---|---|
| 50–360 | 12-14 Oct, 2015 | Kara Strait and Kara Sea | Aethalometer (MSU-CAO: 450, 550, and 650 nm) | Akademik Treshnikov | Popovicheva et al., 2017 |
| 46, 11 | 21 Aug, 2018 | Kara Sea | Quartz fiber filter samples subjected to Aethalometer analysis | Akademik Mstislav Keldysh | Pankratova et al., 2020 |
| 77±17 | Second half of July, 2017 | Norwegian Sea | Quartz fiber filter samples subjected to Aethalometer analysis | Akademik Mstislav Keldysh | Shevchenko et al., 2019 |
| 44±37 | Summar and Autumn of 2007-2020 | Norwegian Sea | MDA Aethalometer or quartz fiber filter samples analyzed by absorption photometers | Akademik Mstislav Keldysh, Akademik Fedorov, Akademik Tryoshnikov, Professor Molchanov, or Professor Multanovsky | Sakerin et al., 2021 |
| 23±11 | 31 Aug-4 Sep, and 13-21 Sep, 2017 | Laptev Sea | Quartz fiber filter samples subjected to Aethalometer analysis | Akademik Mstislav Keldysh | Shevchenko et al., 2019 |
| 6 | 24-25 Aug, 2018 | Laptev Sea | Quartz fiber filter samples subjected to Aethalometer analysis | Akademik Mstislav Keldysh | Pankratova et al., 2020 |
| 2 | 31 Aug-5 Sep, 2018 | Laptev Sea | Quartz fiber filter samples subjected to Aethalometer analysis | Akademik Mstislav Keldysh | Pankratova et al., 2020 |
| 8.3±6.0 | Jun-Oct, 2020 | Central Arctic (>80° N) | Aethalometer (AE33: 880nm) | Polarstern | Boyer et al., 2023 |
| 71±34 | Jan-May, 2020 | Central Arctic (>80° N) | Aethalometer (AE33: 880nm) | Polarstern | Boyer et al., 2023 |

[a] xx±xx indicates mean±standard deviation.

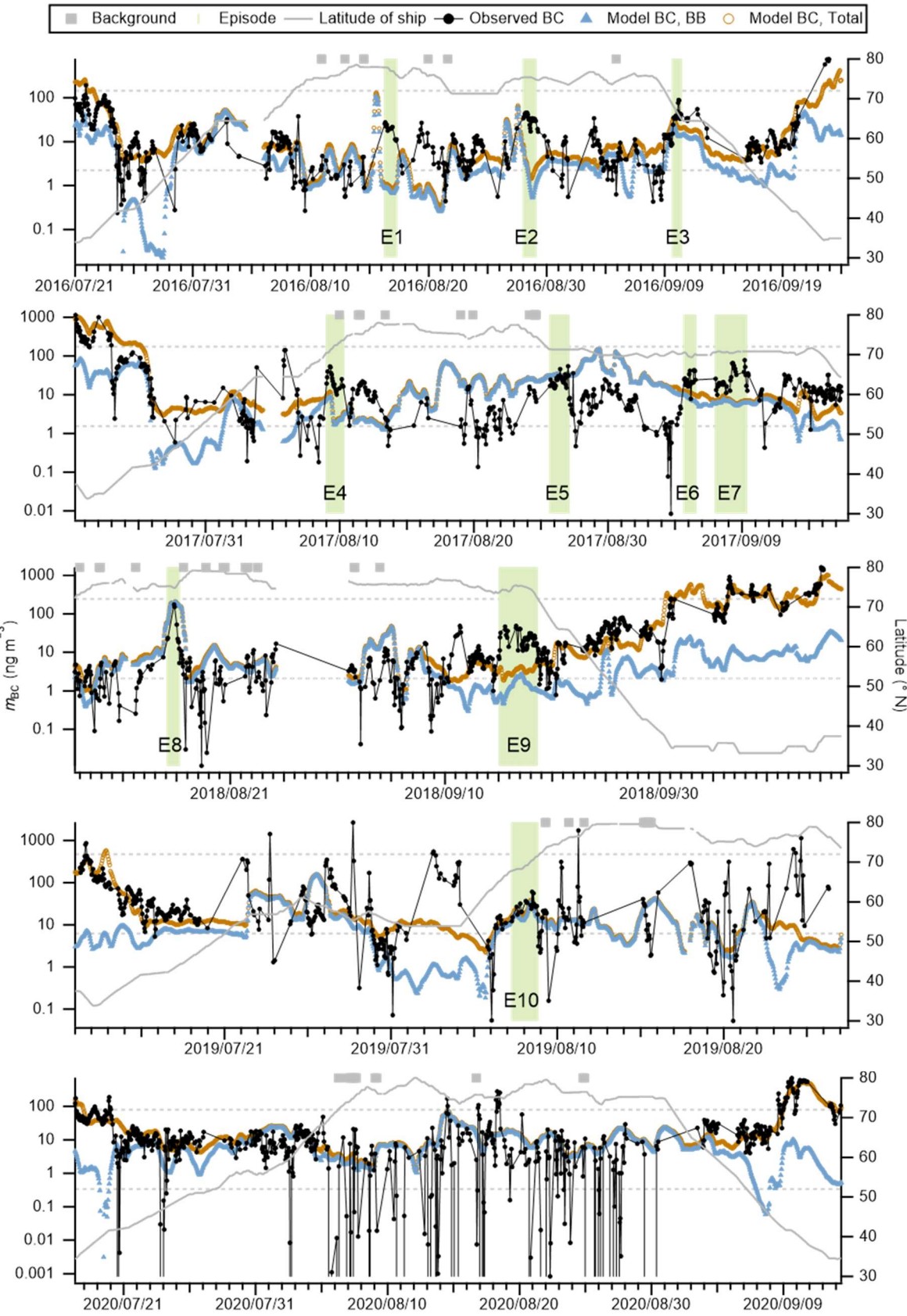

**Figure 3** Time series of mass concentrations of observed BC, model simulated total BC and biomass burning BC, and latitude of ship positions. The Arctic Ocean background periods defined in Sect. 4.2 and 10 high BC episodes (E1 to E10) are also shown. The dashed light gray lines represent latitudes 52° N and 72° N. The $m_{BC}$ presented here is at 1 h time resolution and the data influenced by ship exhaust has been removed. The time series of raw 1-h $m_{BC}$ before removing the influence of ship exhausts are presented in Fig. S6.

## 4.2 Background BC concentration in the western central Arctic Ocean

To evaluate the air quality and climate changes in the Arctic Ocean correctly, it is important to estimate the background BC mass concentration. While finding a situation entirely identical to the preindustrial atmosphere is challenging due to the pervasive influence of anthropogenic activities on even natural events like wildfires (McCarty et al., 2021), examining periods in the Arctic Ocean unaffected by regional transport could offer insights into the preindustrial atmospheric situations. This assumes that the impact of natural terrestrial activities, such as wildfires, on BC in the preindustrial Arctic Ocean atmosphere was likely negligible, recognizing the inherent uncertainties in making such historical assessments.

Many anthropogenic and natural activities can bring BC aerosols to the Arctic Ocean atmosphere. Those activities include industry activities producing large amounts of air pollutants in lower latitude regions that may be transported to the Arctic through long-range transport (Ikeda et al., 2017; Zhu et al., 2020), gas flaring and wildfire frequently occurring in the Arctic regions (Stohl et al., 2013), as well as expanding local activities such as cruise tourism along the Arctic coastal region driven by the warming Arctic climate (AMAP, 2021a). In winter and early spring, the buildup of terrestrial anthropogenic and natural pollutants occurs due to the expansion of the polar dome, which allows for the transport of pollutants from continental regions further south. This buildup, combined with stable atmospheric conditions, can result in monthly mean $m_{BC}$ levels exceeding 100 ng m$^{-3}$ (e.g., Boyer et al., 2023). In summer and early autumn, intense wildfires in the boreal regions can also result in remarkably high $m_{BC}$ levels, as discussed in Sect. 4.1. However, during this period, the $m_{BC}$ in the Arctic Ocean surface layer atmosphere can be extremely low. This is due to changes in transport patterns and wet deposition processes, which efficiently prevent the transport of terrestrial aerosols to the Arctic Ocean (e.g., Bozem et al., 2019; Sierau et al., 2014). Therefore, the summer and early autumn months are considered the most suitable for evaluating the background level of $m_{BC}$ in the Arctic Ocean, with the assumption mentioned previously.

The background periods in the western central Arctic Ocean (>72° N) were determined according to the following criteria: first, for each hour with effective BC data, all three 5-day HYSPLIT back trajectories initiated at starting heights of 10, 500, and 1000 m originated from the Arctic Ocean. Additionally, all 1-min $m_{BC}$ or 5-min $m_{eBC}$ data within that hour were not removed due to ship exhaust according to data

screening criteria described in Sect. 2. The second criterion is to ensure the accuracy of the selected data. As shown in Fig. 3, background periods of 8, 12, 17, 15, and 13 hours were identified for the 2016, 2017, 2018, 2019, and 2020 cruises, respectively. The mean background $m_{BC}$ ($m_{eBC}$) values during the cruises in respective years are presented in Table 1, spanning a broad range from 2 to 14 ng m$^{-3}$. Except for the 2017

cruise, the mean background $m_{BC}$ ($m_{eBC}$) values were lower than their respective overall means and exhibited a positive correlation ($R = 0.98$) with the overall means. The former indicates that the Arctic Ocean could be frequently affected by local or regional BC pollutants. The positive correlation might indicate the accumulation of atmospheric pollutants within the Arctic Ocean planetary boundary layer even in the summer and early autumn months. In the 2017 cruise, the estimated background $m_{BC}$ was higher than the

overall mean, possibly due to residual ship exhaust contamination, despite rigorous data screening procedures (Sect. 2). The overall mean of background $m_{BC}$, calculated from COSMOS $m_{BC}$ at 1 h time resolution over 52 hours, was 7.5 ($\pm$8.5) ng m$^{-3}$. Despite the significantly higher lower detection limit of AE22 used in the 2020 cruise compared to that of COSMOS (Sect. 2), the combined data from both COSMOS $m_{BC}$ and AE22 $m_{eBC}$ for 65 hours resulted in a similar mean background $m_{BC}$ of 7.1 ($\pm$8.2) ng

m$^{-3}$.

To our knowledge, this is the first study that calculated the background concentration of BC in the Arctic marine boundary layer during summer periods. The strong correlation between the estimated mean background $m_{BC}$ values and their respective overall mean values might indicate that the Arctic Ocean atmosphere is readily influenced by long-range transported air pollutants, whose dispersion may be

inhibited by the polar dome. This adds difficulty to the estimation. Nevertheless, the data used for the estimation are mostly limited to within the western Arctic Ocean, and the number of data used for the estimation is small. Future studies based on a larger data size over broader areas in the Arctic Ocean are promising to provide a better estimation of the Arctic Ocean background BC concentration.

## 4.3    Comparisons between observations and model simulations

The time series of GEOS-Chem model simulated total BC mass concentration ($m_{BC,S}$) and that were ascribed to biomass burning sources ($m_{BC,SBB}$) are also presented in Fig. 3, which shows that GEOS-Chem overestimated some low $m_{BC}$ (e.g., during 28 August – 5 September 2017) and underestimated some high $m_{BC}$ (e.g., 18 August 2020). Scatter plots between $m_{BC,S}$ and $m_{BC}$ are presented in Fig. S7. Except for the 2019 cruise, the $R$ values in other years were greater than 0.5. The $R$ for the overall model versus observed

$m_{BC}$ is 0.66. Therefore, GEOS-Chem model can reproduce ($0.66 \times 0.66 =$) 44 % of the temporal and spatial variations of the shipborne $m_{BC}$. The normalized mean biases (NMB) of model simulated from observed $m_{BC}$ in 2017 and 2019 were high and were 102.9 % and −37.4 %, respectively. They were lower in other years, with 16.6, −8.5, and −3.1 % in 2016, 2018, and 2020, respectively. The overall normalized mean

bias was estimated to be 4.6 %. Statistical analysis in NMB showed no distinct spatiotemporal variation characteristics (Fig. S8). Furthermore, the ratio of mean absolute error to the mean of $m_{BC}$ (MAE/Mean) ranged from 0.5 to 1.4 for individual cruises, with an overall estimate of 0.8 for all cruises (Fig. S7). This suggests that the model can reproduce observed data with an average relative uncertainty of less than 1.4.

GEOS-Chem failed to reproduce almost all the high BC spikes observed in the Arctic Ocean in 2019. As discussed in Sect. 4.1, the high spikes in 2019 were likely caused by intensive wildfires in the Arctic especially Eastern Siberia (Antokhina et al., 2023) and Alaska (Bhatt et al., 2021). Therefore, we can infer that the less accounting of wildfires in the boreal regions by the GFED4s biomass burning inventory used in this study might be the main reason for the poor reproduction of observed BC during the 2019 cruise by

GEOS-Chem (Pan et al., 2020) considering that the transport path of BC from the boreal regions to the Arctic Ocean is mainly through the lower to middle atmosphere as indicated by the analyses in Sect. 4.4 and previous studies (e.g., Ikeda et al., 2017; AMAP, 2021b). Thus, it is necessary to improve the estimation of biomass burning emissions in the boreal regions. However, the influence of possible uncertainties in the transport regime of the GEOS-Chem model (e.g., overestimation of wet deposition) in reproducing the

peaks observed during the 2019 Arctic cruise cannot be ruled out. In addition, Fig. S7 shows systematical overestimation of model $m_{BC}$ in the region of lower than 1 ng m$^{-3}$. Similar overestimation was also found in Whaley et al. (2022), which was possibly caused by the coarse resolution of the GEOS-Chem model, making it unable to accurately simulate such low BC mass concentrations.

## 4.4   Sources of High BC episodes

Statistical analyses of the GEOS-Chem simulated biomass burning BC to total BC ratio ($m_{BC,SBB}/m_{BC,S}$) are presented in Fig. 2b, which indicate that BCbb accounted for more of the observed BC mass concentration in higher than in lower latitude regions. In the Arctic Ocean (i.e., north of 72° N), BCbb contributed on average 67–92 % of total BC observed along the cruise tracks. In the marginal Arctic Sea regions (i.e., between 52 and 72° N), $m_{BC,SBB}/m_{BC,S}$ was estimated to be 62–74 % except in the 2018 cruise, where it was

estimated to be 16 % (Fig. 2b). These results indicate that most of the observed BC in the Arctic during summer and Autumn were from biomass burning sources. This aligns with previous model studies indicating that during summer, the transport efficiency of low latitude anthropogenic BC to the Arctic was low, and biomass burning BC contributed more than 63 % to the surface BC in the Arctic (Ikeda et al., 2017; Zhu et al., 2020).

Elevated BC mass concentration periods were observed in almost every Arctic cruise (Fig. 3). To characterize the sources of the high concentrations of BC in the Arctic Ocean and the marginal seas (north of 65° N), we identified periods when the 1-h $m_{BC}$ exceeded 10 ng m$^{-3}$. From these periods, we further selected those lasting 18 h or longer and the mean of valid 1-h $m_{BC}$ during the selected period was not less

than 20 ng m$^{-3}$. This process allowed us to identify and refine 10 high BC episodes (Figs. 1, 3, S1, and S6, Table 3). Episode 1 (abbreviated as E1, same for other episodes) and E8 were observed in the Arctic Ocean; E2 was observed in the East Siberian Sea; E3 was observed on the way from the Chukchi Sea to the Bering Strait; E4 and E10 were observed in the Chukchi Sea; E5, E6, and E7 were observed in the Beaufort Sea near the Alaska coast; and E9 was observed on the way from the East Siberian Sea to the Chukchi Sea. Table 3 presents the time and space range details and observed and model simulated mean BC mass concentrations during the 10 episodes. According to GEOS-Chem model simulation results, except for E9 which occurred in 2018, biomass burning contributed more than 69 % of the observed BC during all the other episodes (Table 3). Note that despite substantial normalized mean biases in model simulations compared to observed $m_{BC}$ for these episodes, ranging from −95 % to 178 %, we consider it reasonable to estimate the contribution of biomass burning to the total BC based on the model results. This is attributed to the pervasive dominance of biomass burning BC north of 65° N, where all episodes were identified (Fig. 3). The estimate is further supported by the uncertainty analysis, involving shifting the episode period back or forward by 18 hours while maintaining its length, which revealed changes of no more than 10 % in the modeled biomass burning to total BC ratio for most episodes. Additionally, the temporal and spatial variations of E3, E8, and E10 were well reproduced by GEOS-Chem model, showing nearly simultaneous peaks in observed and model data during these episodes (Fig. 3). Therefore, in the following sections, the sources and transports of BC during E3, E8, and E10 are elaborated based on GEOS-Chem model, with findings further corroborated by the HYSPLIT back trajectory model.

**Table 3** Time and space ranges and observed and simulated BC mass concentrations for the 10 episodes.

| Episodes | Start time | Duration (h) | | Latitude (°) | Longitude (°) | Mean $m_{BC}$ (ng m$^{-3}$) | | $m_{BC,SBB}/m_{BC,S}$[c] (%) |
|---|---|---|---|---|---|---|---|---|
| | | Total[a] | Valid[b] | | | Observed | Model, total | |
| E1 | 16 Aug 2016, 06:00 | 25 | 14 | +76.00– +77.88 | −175.89– −174.78 | 20 | 1.0 | 82 |
| E2 | 28 Aug 2016, 00:00 | 26 | 25 | +74.89– +75.40 | +170.93– +171.86 | 34 | 2.9 | 72 |
| E3 | 9 Sep 2016, 16:00 | 18 | 18 | +65.05– +68.48 | −168.48– −168.41 | 44 | 25 | 69 |
| E4 | 9 Aug 2017, 00:00 | 31 | 19 | +70.49– +73.58 | −168.71– −168.28 | 25 | 5.6 | 82 |
| E5 | 25 Aug 2017, 16:00 | 34 | 32 | +71.32– +71.33 | −156.88– −156.79 | 25 | 33 | 97 |
| E6 | 4 Sep 2017, 16:00 | 21 | 17 | +69.34– +70.57 | −139.02– −138.21 | 26 | 11 | 73 |
| E7 | 7 Sep 2017, 00:00 | 56 | 30 | +70.38– +70.81 | −140.02– −135.31 | 32 | 7.3 | 87 |
| E8 | 15 Aug 2018, 03:00 | 27 | 9 | +74.80– +76.26 | −171.97– −166.32 | 55 | 154 | 98 |
| E9 | 15 Sep 2018, 02:00 | 84 | 50 | +72.52– +75.50 | +167.84– −168.36 | 25 | 3.5 | 41 |

| | | | | | | | |
|---|---|---|---|---|---|---|---|
| **E10** | 7 Aug 2019, 06:00 | 38 | 21 | +67.80–+71.50 | −168.67–−167.12 | 29 | 23 | 86 |

[a] The total duration of each episode.

[b] The number of hours with valid 1-h BC mass concentration data in each episode.

[c] GEOS-Chem simulated biomass burning to total BC mass concentration ratio.

### 4.4.1  Episode 3

Episode 3 was measured during 9 September 2016, 16:00 – 10 September 2016, 10:00 UTC. The mean $m_{BC}$ is 44 ng m$^{-3}$ and BC from biomass burning was estimated to contribute 69 % of the total BC (Table 3). Figure 4 presents the surface distribution of BCbb and the surface winds before this episode. It suggests that the biomass burning occurred on boarder of Chukotka Autonomous Okrug (CAO) and Kamchatka Krai (KamK) was likely the main source of this episode. Southwest winds have brought the biomass burning BC containing airmass from the source region to the ship positions. Figure 4 indicates that biomass burning contributed 80 % of the BC mass concentration at this source region. GFED4s data and back trajectories (Fig. S9) also indicate that biomass burning occurred on boarder of CAO and KamK was likely the main source of the observed high BC mass concentration during E3. The longitude-height cross sections of BCbb also presented in Fig. 4 suggests that the height of the transport path of BCbb was constrained to >700 hPa (i.e., <~3 km) and little contribution of subsidence BC from upper atmosphere had contributed to E3. This is also supported by the height distributions of back trajectories (Fig. S9), which indicate that the observed airmasses were transported to the ship position within 2.5 km above the ground level. Compared with BCbb, the contribution of anthropogenic BC to the observed high BC mass concentrations in E3 was relatively small through either surface level or above-ground transports (Fig. S10).

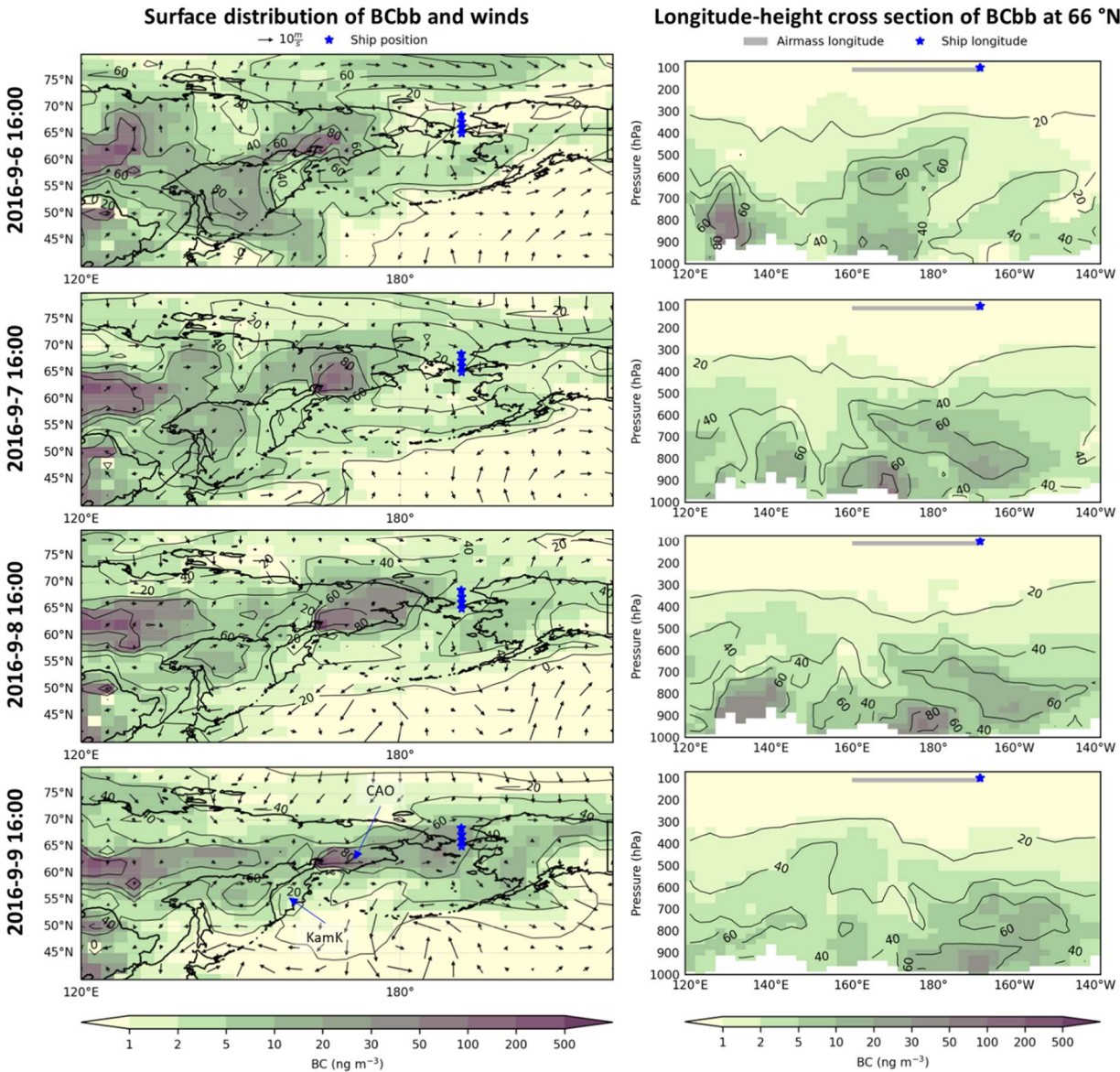

**Figure 4** Simulated biomass burning BC (BCbb, color image) surface distributions (left) and longitude-pressure cross sections at 66° N (right) before Episode 3. Superimposed on the left panels are surface winds and the ship positions. Superimposed on the right panels are the ship longitude positions and the possible transport region of BC-containing air masses related with Episode 3. The latter was inferred from GEOSChem model (left) and back trajectories (Fig. S9). On both panels, the contour plot represents the simulated biomass burning BC to total BC ratio (%). In the bottom left panel, CAO-Chukotka Autonomous Okrug and KamK-Kamchatka Krai.

## 4.4.2 Episode 8

Episode 8 was measured during 15 August 2018, 3:00 – 16 August 2018, 6:00 (Fig. 3). The mean $m_{BC}$ is 55 ng m$^{-3}$ and BC from biomass burning was estimated to contribute 98 % of the total BC (Table 3). Note

that although there are only nine valid 1-h $m_{BC}$ during E8, following analyses on gaseous species and GEOS-Chem and back trajectory model simulations indicate that E8 is part of a prominent transport event of Siberia biomass burning airmasses to the Arctic Ocean. Figure 5 presents the surface distribution of biomass burning BC and winds before, during, and after Episode 8. Biomass burning airmasses from Krasnoyarsk Krai (KraK) and the Republic of Sakha (Sakha) were transported northwards and northeastwards to the Siberia Arctic and then spread eastwards by the westerly from 13 to 14 August. Further, northwest winds blew the biomass burning BC containing airmasses to the ship positions on 15 August. Figure 5 indicates that biomass burning contributed more than 80 % of BC mass concentration in the transported airmasses. These transport paths are also supported by GFED4s data and back trajectory analyses (Fig. S11). Figure 5 also shows that the biomass burning BC containing airmass was blown away from the ship later by northerly winds. Although the height distribution of back trajectories presented in Fig. S11 showed that the observed air masses during E8 were transported to the ship position mainly under 2 km above the ground, longitude-height distributions of BCbb presented in Fig. 5 indicate that the transport of biomass burning BC containing airmasses from the source regions to above the ship position was mainly through lower to middle atmosphere. Although the contribution of anthropogenic BC to the observed BC in Episode 8 was very small, surface level concentration distribution and longitude-height cross sections (Fig. S12) show that they followed similar transport paths to the ship positions as the biomass burning BC.

Figure 6 presents the time series of the atmospheric mixing ratios of CO, $CH_4$, $CO_2$, and $O_3$, as well as observed and model simulated BC mass concentrations during the 2018 shipborne observation when CO, $CH_4$, and $CO_2$ data were obtained. During Episode 8, the mixing ratios of CO and $CH_4$ increased whereas those of $CO_2$ and $O_3$ were not or even slightly decreased. Similar phenomena have been reported in previous studies in the lower atmosphere over Siberia (Paris et al., 2010). The increased CO and $CH_4$ is consistent with the observation of biomass burning plumes possibly related with smoldering combustion conditions (Andreae et al., 1994). The slight decrease in $CO_2$ is possibly due to uptake by intact high latitude vegetation during the polar daylight period before transporting to the Arctic Ocean (Paris et al., 2010) as well as smoldering combustion conditions, producing much more CO than $CO_2$. The former is consistent with the fact that in Siberia planetary boundary layer and free troposphere $CO_2$ concentrations are at the minimum in July to August (Sasakawa et al., 2013). The no increase or slight decrease of $O_3$ was possibly caused by less active photochemistry in the fire plumes, in particular, at the northern high-latitude (Tanimoto et al., 2000) and/or surface deposition (Text S1). Over Siberia, the $O_3$ formed in biomass burning plumes probably was lost greatly due to deposition to the forest canopy before being transported out of the Siberia terrestrial to the Arctic Ocean (Chin et al., 1994; Paris et al., 2010) so that the observed $O_3$ concentration in the plumes were lower than that of the Arctic Ocean background. Scatter plots between $m_{BC}$ versus CO, $O_3$ versus CO, CO versus $CO_2$, and $CH_4$ versus $CO_2$ are presented in Fig. 7, where most of the data points during E8 are

significantly different from the others. The reduced major axis regression between $O_3$ and CO during period having not been influenced by Episode 8 airmasses resulted in a slope of 0.39, which is similar to that derived from the MOSAiC observation in the central Arctic during the same season of 2020 (Fig. S13; Angot et al., 2022 and references therein). The spatial distribution of the atmospheric mixing ratios of $O_3$,

5   CO, $CO_2$, and $CH_4$, and the $m_{BC}/\Delta CO$ (i.e., the enhancement ratio of BC to CO; here, $\Delta CO$ is the increase in CO relative to baseline, see the caption of Fig. 8 for more details; note that in order to ensure that there were sufficient data to characterize the spatiotemporal changes in $m_{BC}/\Delta CO$, the background of $m_{BC}$ was not subtracted) and $CO/CO_2$ ratios are presented in Fig. 8. Distinctive features such as increases of CO, $CH_4$, $m_{BC}/\Delta CO$ and $CO/CO_2$ ratios and decreases of $CO_2$ and $O_3$ during E8 can be clearly observed. In

10   addition, the median $m_{BC}/\Delta CO$ of less than 1 ng m$^{-3}$ ppb$^{-1}$ is near to those reported in Taketani et al. (2022), which might have been affected by wet removal of BC during transport processes or smoldering combustion conditions.

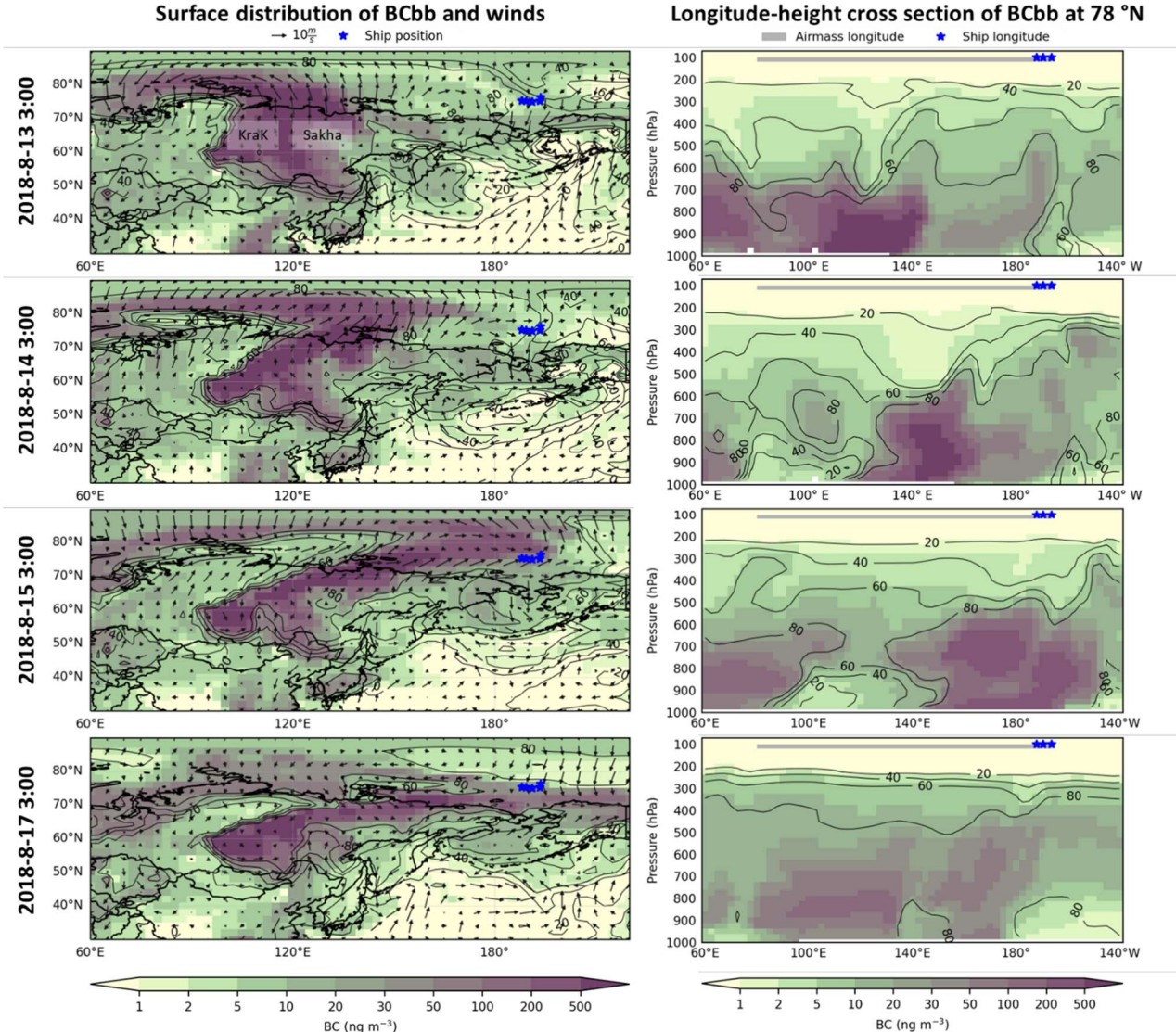

**Figure 5** Simulated biomass burning BC (BCbb, color image) surface distributions (left) and longitude-pressure cross sections at 78° N (right) before to right after Episode 8. Superimposed on the left panels are surface winds and the ship positions. Superimposed on the right panels are the ship longitude positions and the possible transport region of BC-containing air masses related with Episode 8. The latter was inferred from GEOSChem model (left) and back trajectories (Fig. S11). On both panels, the contour plot represents the simulated biomass burning BC to total BC ratio (%). In the upper left panel, KraK- Krasnoyarsk Krai and Sakha- the Republic of Sakha.

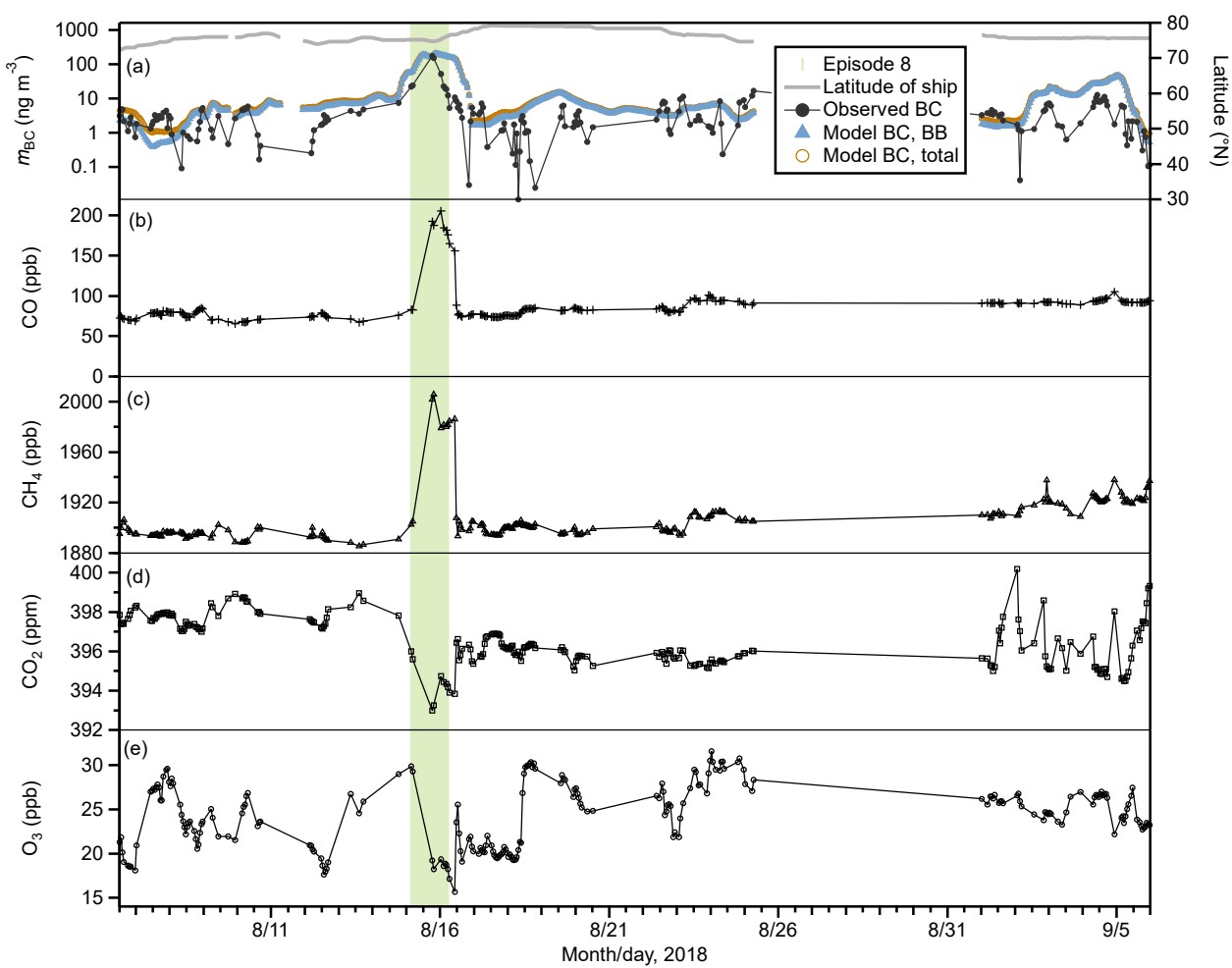

**Figure 6** Time series of (a) observed BC, model simulated total BC and biomass burning BC, and latitude of ship positions, (b) CO, (c) $CH_4$, (d) $CO_2$, and (e) $O_3$ during the 2018 shipborne observation. Bar shade indicates the Episode 8 period.

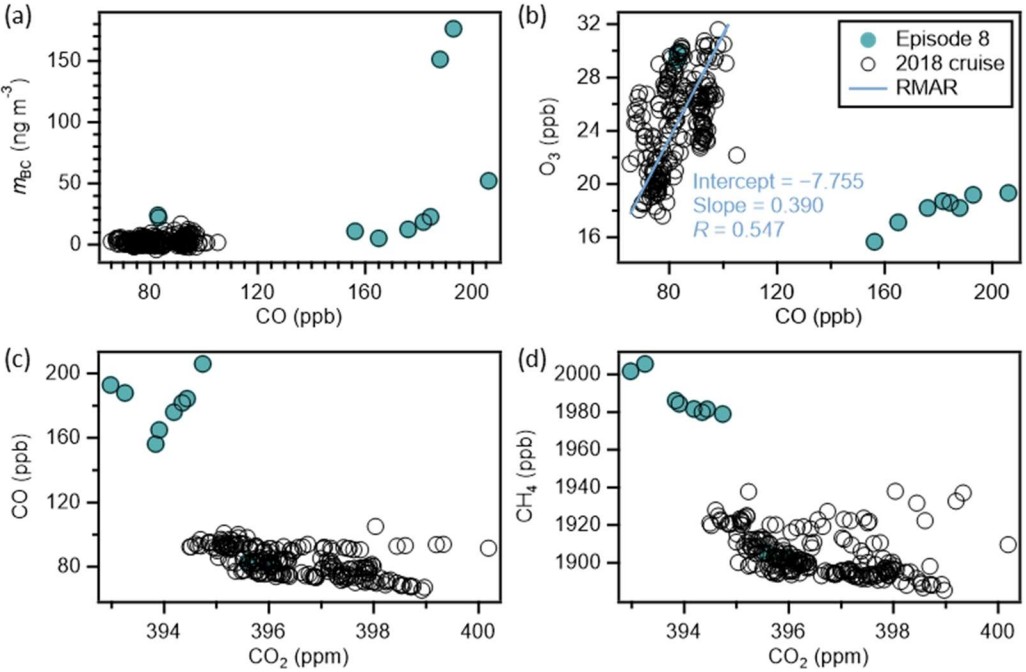

**Figure 7** Scatter plots of (a) $m_{BC}$ versus CO, (b) $O_3$ versus CO, (c) CO versus $CO_2$, and (d) $CH_4$ versus $CO_2$ during the cruise in the Arctic Ocean in 2018 and Episode 8. In panel (b), the line represents the reduced major axis regression (RMAR) for data having not been influenced by Episode 8 airmasses: the intercept, slope, and correlation coefficient are also presented.

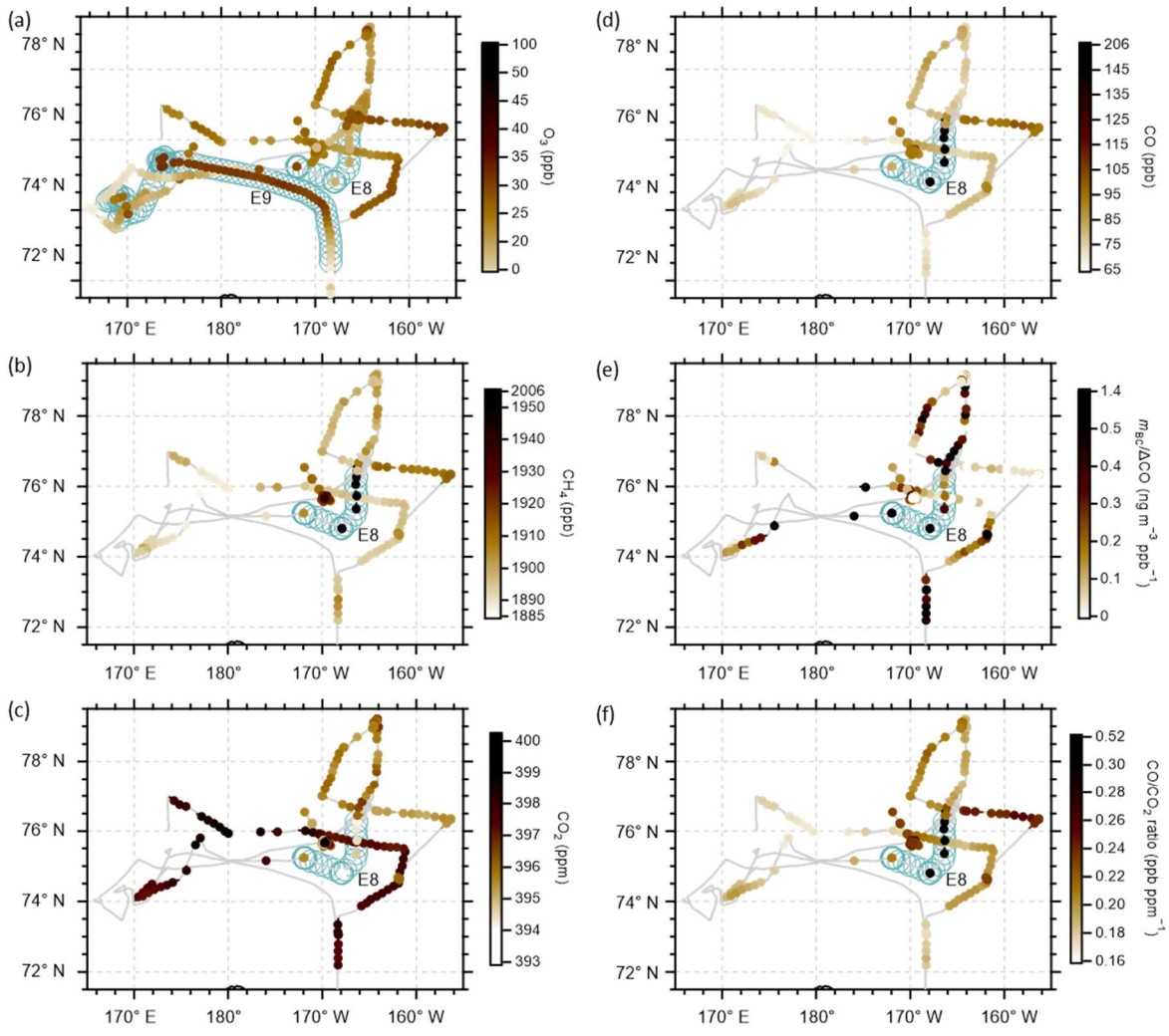

**Figure 8** Surface distributions of $O_3$ (a), $CH_4$ (b), $CO_2$ (c), and CO (d) mixing ratios, and $m_{BC}/\Delta CO$ (e) and $CO/CO_2$ (f) ratios along the ship track during part of the 2018 cruise in the Arctic Ocean. In each panel, the grey line represents the cruise track; the filled color markers superimposed on the track indicates the respective observed (a, b, c, and d) or derived (e and f) parameters, which are at 1 h time resolution and screened to remove the influence of ship exhausts; and the open circles represent the ship positions during Episodes 8 (a-f) and 9 (a). Note that valid $CH_4$, CO, and $CO_2$ data are only available for a limited time (Fig. 6). For the derivation of $\Delta CO$, the baseline of CO is defined as the minimum 1-h CO data; and $m_{BC}/\Delta CO$ ratio was calculated only when $\Delta CO$ was higher than 4 ppb.

### 4.4.3 Episode 10

Episode 10 was measured during 7 August 2019, 6:00 – 8 August 2019, 20:00 UTC (Fig. 3). The mean $m_{BC}$ is 29 ng m$^{-3}$ and BC from biomass burning was estimated to contribute 86 % of the total BC (Table 3). Figure 9 presents the surface distribution of biomass burning BC and surface winds before and during Episode 10. Although no obvious fire spot was observed on boarders among Magadan Oblast and Chukotka

Autonomous Okrug and Kamchatka Krai (abbreviated as MCK boarders, Figs. 9 and S10), GEOS-Chem simulations showed high concentration of biomass burning BC at MCK boarders on 6 August 2019 (Fig. 9), which was then transported to the ship position by weak northeastward winds. Longitude-pressure cross sections of BCbb presented in the right panels of Fig. 9 suggest that the high BCbb occurred at MCK boarders (150–170° E) was likely subsidence from upper atmosphere. Surface BCbb distributions (Fig. 9) and GFED4s map (Fig. S14) show that intensive biomass burning occurred in Krasnoyarsk Krai (KraK), Irkutsk Oblast (IrO), and the Republic of Sakha (Sakha) areas (90–150° E) before and during Episode 10. This is consistent with Antokhina et al. (2023), which reported intensive fire activities during 3 July to 12 August 2019 in Siberia (95–120° E). The highly BCbb containing airmasses from these intensive fires advected up to 4 km (i.e., ~600 hPa) and transported to the ship position mainly through the lower to middle atmospheres (Fig. 9right). Figures S15 and S16 show the horizontal BCbb distribution and wind fields at about 800 and 600 hPa, respectively. Both figures indicate that subsidence of BCbb containing airmasses occurred at MCK boarders, and the rest of BCbb containing airmasses were transported to the Arctic above the ship positions through much stronger than surface southwest winds, which are consistent with the longitude-pressure cross section of BCbb (Fig. 9). In addition, the height distribution of back trajectories also showed that more than a third airmasses originated from an altitude higher than 2 km (Fig. S14). Contour plots superimposed on each figure (Figs. 9 and S15-16) indicate that biomass burning BC contributed to more than 80 % of the BC transported to the ship position. Surface distributions and longitude-height distributions (Fig. S17) of anthropogenic BC show that it contributed little to the observed BC in Episode 10.

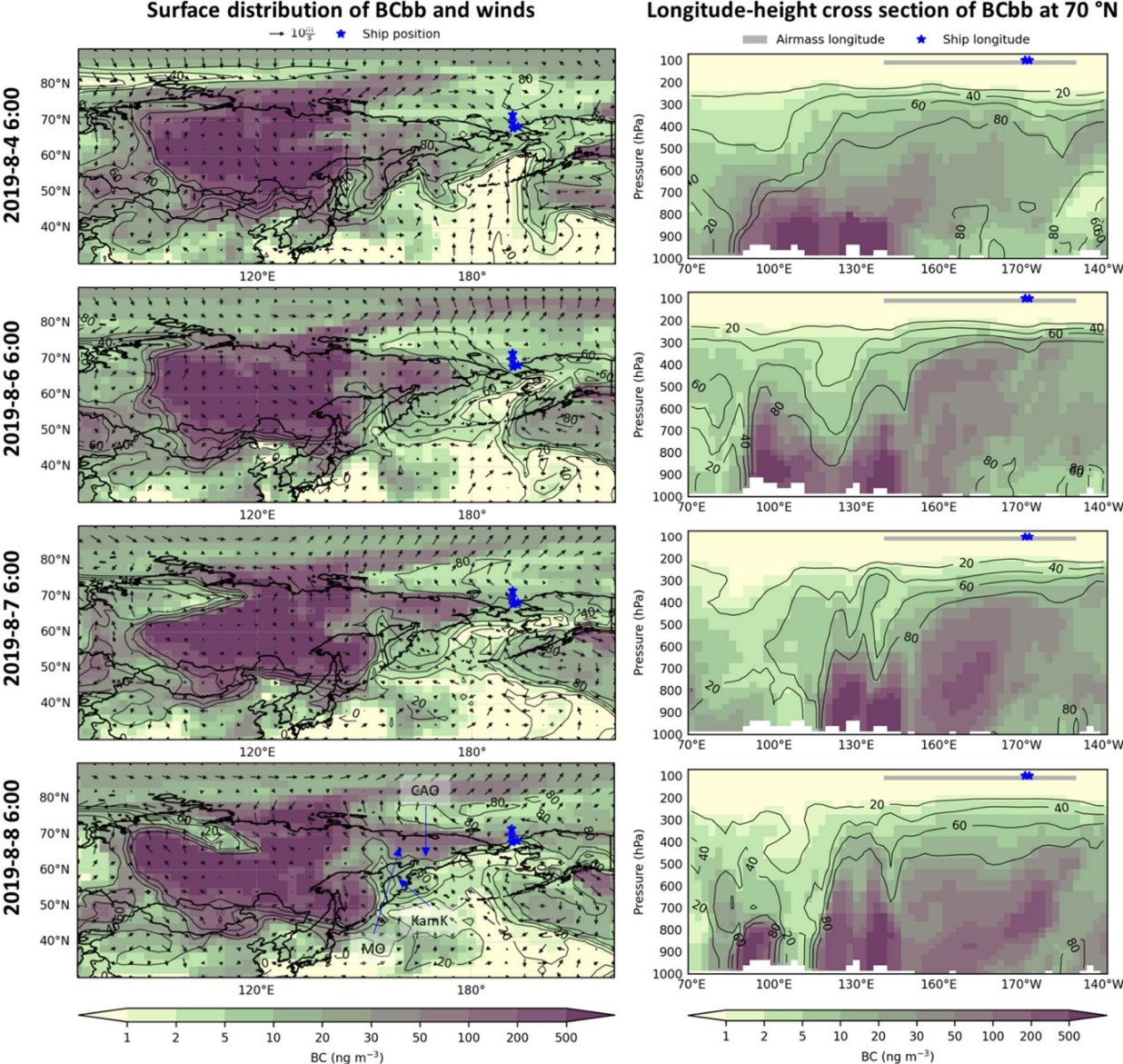

**Figure 9** Simulated biomass burning BC (BCbb, color image) surface distributions (left panel) and longitude-pressure cross sections at 70° N (right panel) before and during Episode 10. Superimposed on the left panels are surface winds and the ship positions. Superimposed on the right panels are the ship longitude positions and the possible surface transport region of BC-containing air masses related with Episode 10. The latter was inferred from GEOSChem model (left) and back trajectories (Fig. S14). On both panels, contour plots represent the simulated biomass burning BC to total BC ratio (%). In the lower left panel, MO- Magadan Oblast, CAO- Chukotka Autonomous Okrug, and KamK- Kamchatka Krai.

# 5 Summary and conclusions

The mass concentration of black carbon aerosols was measured in the Arctic Ocean, encompassing the western Arctic Ocean and part of the East Siberian Sea, as well as the North Pacific Ocean. The measurements were conducted using COSMOS and an AE-22 Aethalometer on board icebreaker R/V *Araon* during summer and autumn 2016–2020. Relatively low levels of $m_{BC}$ were observed at higher latitude regions. In the western Arctic Ocean (>72° N), the overall mean (± 1 standard deviation) of 1-h $m_{BC}$ during the cruises in 2016, 2017, 2018, 2019, and 2020 were 10 (±11), 6.6 (±6.7), 7.8 (±15), 73 (±210), and 14 (±35) ng m$^{-3}$, respectively. The estimated background $m_{BC}$ concentrations in respective years show a strong positive correlation with those mean values, indicating potential accumulation of atmospheric pollutants within the Arctic Ocean planetary boundary layer even in the summer and early autumn months. The overall mean of the background $m_{BC}$ across all five cruises was estimated to be 7.1 (±8.2) ng m$^{-3}$. In the western Arctic Ocean and the Bering Sea (>52° N), the year-to-year variation of $m_{BC}$ was not significant, except for the 2019 cruise, which observed much higher and more frequent elevated $m_{BC}$ compared to other years. This increase was likely attributed to more frequent biomass burning in the Arctic region in 2019. We identified 10 high BC episodes north of 65° N based on the observational data. Significant but irregular interannual variability in $m_{BC}$ was observed in the North Pacific Ocean (south of 52° N).

Tagged tracer simulations of BC using a global chemistry transport model (GEOS-Chem) were applied for the interpretation of the sources and transport paths of the observed BC. The model's relative uncertainty, estimated based on the observed $m_{BC}$, was less than 1.4. Additionally, the model was estimated to reproduce 44 % of the temporal and spatial variations of $m_{BC}$. GEOS-Chem analyses indicate that biomass burning composed the largest contribution to the observed BC along the ship tracks in the Arctic Ocean (67–92 %) and most high BC episodes (41–98 %). GEOS-Chem also revealed that transport paths of biomass burning BC from Siberian area to the Arctic could occur near-surface and/or through the lower to middle atmosphere. However, GEOS-Chem failed to accurately replicate the frequently observed high BC spikes in the Arctic during the 2019 cruise, which were attributed to the influx of biomass burning airmasses. This suggests the need for improvements in biomass burning emission inventories, especially considering the ongoing increase in wildfires during the boreal summer in a warming climate. Nevertheless, it cannot be ruled out that uncertainties in the BC transport regimes used in GEOS-Chem also contributed to the simulation discrepancies.

This study provides crucial datasets on BC mass concentrations and the mixing ratios of $O_3$, $CH_4$, CO, and $CO_2$ in the western Arctic Ocean regions during summer and autumn. Our results also highlight the significant impact of boreal fires on the observed Arctic BC mass during summer and early autumn months,

consistent with previous modelling and observational studies (e.g., Zhu et al., 2020; Popovicheva et al., 2022). These results are valuable for model validation, predicting Arctic climate change, and guiding air quality research in the Arctic Ocean. In addition, due to rapid changes in temperature, precipitation, snow cover, sea and land ice, permafrost, and extreme events occurring in the Arctic (AMAP, 2021a), the sources, transport pathways, and climate forcing effects of BC are thought to be changing in the Arctic. Therefore, further studies on the spatial-temporal distributions, background concentrations of BC in the Arctic marine boundary layer, and the impact of boreal fires as well as other natural and anthropogenic sources on Arctic Ocean atmospheric BC are required to clearly understand the feedback of atmospheric BC in the rapidly changing Arctic Ocean.

# Appendix A: Statistics of gaseous species

**Table A1**: Statistics of the observed concentrations of gaseous species during shipborne measurements in 2017 and 2018.

| Year | | 2017 | 2018 | | | | | | |
|---|---|---|---|---|---|---|---|---|---|
| **Species** | | $O_3$ (ppb) | $O_3$ (ppb) | $CH_4$ (ppb) | CO (ppb) | $CO_2$ (ppm) | $CO/CO_2$ ratio (ppb ppm$^{-1}$) | $m_{BC}/\Delta CO$ ratio (ng m$^{-3}$ ppb$^{-1}$) |
| **North of 72° N** | Median | 23.2 | 24.2 | 1900.7 | 82.2 | 396.21 | 0.208 | 0.119 |
| | Mean | 24.3 | 23.6 | 1906.8 | 86.4 | 396.46 | 0.218 | 0.172 |
| | STD | 3.6 | 5.0 | 19.1 | 20.3 | 1.23 | 0.052 | 0.238 |
| **Between 52 and 72° N** | Median | 25.8 | 24.8 | - | - | - | - | - |
| | Mean | 25.1 | 24.1 | - | - | - | - | - |
| | STD | 6.1 | 6.0 | - | - | - | - | - |
| **South of 52° N** | Median | 38.9 | 38.8 | - | - | - | - | - |
| | Mean | 38.0 | 43.2 | - | - | - | - | - |
| | STD | 12.9 | 14.3 | - | - | - | - | - |
| **Whole cruise** | Median | 25.1 | 26.8 | - | - | - | - | - |
| | Mean | 27.2 | 29.3 | - | - | - | - | - |
| | STD | 8.9 | 12.5 | - | - | - | - | - |

Note: STD, standard deviation; -, no available data.

## Data availability

The dataset containing $m_{BC}$ (ng m$^{-3}$), ship latitude and longitude, relative wind direction (RWD), relative wind speed (RWS, m s$^{-1}$), CH4 (ppb), CO (ppb), and CO2 (ppm) used in this publication is available online https://doi.org/10.17595/20240502.001 (Deng et al., 2024).

## Author contributions

HT, SK, and JJ designed the experiment and did the shipborne observations with contributions from SO, YJY, EJY, and SHK; KI did the model simulations; YD analyzed the observation and model data with

contributions from HT and KI; YD made the manuscript with contributions from HT and KI; and all authors contributed to the revisions of the manuscript.

## Competing interests

The authors declare no competing interests.

## Acknowledgements

We thank Ms. Kimiko Suto for her support in gaseous species data analysis. We also thank Dr. Sho Ohata for his valuable suggestions on COSMOS data analysis. We acknowledge the NOAA Air Resources Laboratory (ARL) for providing the HYSPLIT transport and dispersion model. The Aethalometer measurements were supported by a National Research Foundation of Korea Grant from the Korean Government (MSIT; the Ministry of Science and ICT) NRF-2021M1A5A1065425 (KOPRI-PN24011). This research was supported by the Korea Institute of Marine Science & Technology (KIMST) funded by the Ministry of Oceans and Fisheries, Korea (grant no. 20210605, Korea-Arctic Ocean Warming and Response of Ecosystem, KOPRI). Financial support to NIES was given by the Environmental Research and Technology Development Fund (ERTDF) from the Ministry of the Environment, Japan (grant no. 2-2201: JPMEERF20222001), and by JSPS KAKENHI grant number JP21K12216 and JP17KK0016.

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
