# Peer review of "Measurement report: Shipborne observations of black carbon aerosols in the western Arctic Ocean during summer and autumn 2016–2020: impact of boreal fires"

_EGUsphere, 2023_

## Author Comment (AC3)

**Response to Anonymous Referee #1**

We appreciate the invaluable comments. Our answers to the comments are provided below. The reviewer comments are written in italics.

*The manuscript presents BC data, based on absorption measurements from 5 Arctic ship-based campaigns (2016-2020). It is a nice data-set, and considering the scarcity of available data from this remote area, very valuable. I would recommend the paper for publication in ACP, however have some comments that are needed to be addressed before.*

*Both the AE33 and COSMOS are instruments, that measure absorption coefficients. It is already challenging and makes different corrections necessary to retrieve the correct absorption coefficient from filter-based measurements. The absorption coefficients are then used to estimate the value of the BC mass concentrations. Reading the paper, one could think that the instrument would measure directly the BC mass, or at least it would be a trivial or straight-forward thing to get the BC concentrations correctly from filter-based absorption measurements. Only very few details are given according to the measurement uncertainty and possible corrections and assumptions. I would like to read a more thorough discussion/introduction about it, how well these instruments can be used for direct BC mass measurements. The manuscript from Backmann et al., 2017 (https://doi.org/10.5194/amt-10-5039-2017) deals with the uncertainties of Aeth measurements from the Arctic and proposes a region-specific correction factor. I am not even completely sure, how and if the aethalometer data was corrected, but this would be maybe an option to consider.*

Reply> Thank you for the comments. First of all, the model of the Aethalometer which was deployed during the cruise is AE22, not AE33. This mistake is corrected in the revised manuscript. We sincerely apologize for any inconvenience caused by this mistake.

Another major change is that we successfully obtained AE22 data for the 2017 and 2018 cruises, which were not available in the preprint. Therefore, we can compare the two instruments, which led our analysis to focus more on the COSMOS data because of the larger uncertainty in the AE22 data.

We used the manufacturer recommended standard BC mass concentration data from COSMOS and AE22 throughout the analysis. In an effort to give readers more information about the measurements, we modified the text to include the MAC values, the uncertainties, and other specific features of each instrument, and the comparison between the two instruments. With this, paragraph two in Sect. 2 of the preprint has been expanded to three paragraphs in the revised manuscript. That is, the original expression "A continuous soot monitoring system (COSMOS, model 3130, KANOMAX, Japan) and an Aethalometer (model AE33,

Magee Scientific Co., USA) were used in the cruises in 2016–2019 and in 2020, respectively, to measure the mass concentrations of BC aerosols. Both instruments use light absorption methods. Therefore, the obtained mass concentration of BC is equivalent black carbon (eBC, Petzold et al., 2013). Nonetheless, BC will be used throughout the manuscript if not specifically mentioned. COSMOS monitors changes in transmittance of 565 nm wavelength LED light across an automatically advancing quartz fiber filter tape. To achieve measurements with high sensitivity and a lower detectable light absorption coefficient, COSMOS uses a double-convex lens and optical bundle pipes to maintain high light intensity and signal data are obtained at 1000 Hz. The data integration time was set to 1 min, which was then averaged to 1 h for further analysis. In addition, its sampling flow rate (0.9 L min$^{-1}$) and optical unit temperature were actively controlled. The inlet line for COSMOS was heated to 400 °C to effectively volatilize non-refractory aerosol components that were internally mixed with BC. The lowest detection limit of COSMOS at 1 min time resolution is 50 ng m$^{-3}$ and at 1 h time resolution is 1 ng m$^{-3}$ (Ohata et al., 2019). Aethalometer uses the absorption of light at a wavelength of 880 nm by the ambient aerosols collected on a Pallflex Teflon-coated glass fiber (TFE) filter tape to determine the BC concentration. The flow rate was set to 5 L min$^{-1}$. The data integration time was set to 5 min, which was then averaged to 1 h for further analysis. The default mass absorption cross section value of 7.77 m$^2$ g$^{-1}$ and internal multiple scattering correction factor of 1.57 were applied (Drinovec et al., 2015). The lower detection limit of Aethalometer at 1 h time resolution is 5 ng m$^{-3}$. It is noted that the default parameter settings of the aethalometer may cause the obtained BC mass concentrations to be twice the actual values (Laing et al., 2020; Asmi et al., 2021)." has been modified to

"A continuous soot monitoring system (COSMOS, model 3130, KANOMAX, Japan) and an Aethalometer (model AE22, serial number 1057:1010, Magee Scientific Co., USA) were used during the cruises to measure the mass concentrations of BC aerosols. Whereas both instruments use light absorption methods, COSMOS was equipped with a 400 °C heated inlet line. This feature effectively eliminated interference from volatile non-refractory aerosol chemical species internally mixed with BC, ensuring a high accuracy of $m_{BC}$ measurement. This aspect has been critically assessed in previous studies (Ohata et al., 2019; Sinha et al., 2017). Consequently, COSMOS measurements differ from traditional light absorption methods, where the mass concentration of BC is referred to as equivalent BC (eBC, Petzold et al., 2013). Therefore, instead of using eBC, the term BC can be used for COSMOS data in a general sense (Ohata et al., 2019). Henceforth, when comparing data from the two different instruments, we will use $m_{eBC}$ to represent the BC mass concentration measured with the Aethalometer during the 2017, 2018, and 2020 cruises, and $m_{BC}$ (COSMOS) to represent the BC mass concentration measured with COSMOS during the 2016–2019 cruises. Otherwise, BC mass concentration is denoted as $m_{BC}$ for simplicity.

COSMOS monitors changes in transmittance of 565 nm wavelength LED light across an automatically advancing quartz fiber filter tape. To achieve measurements with high sensitivity and a lower detectable light absorption coefficient, COSMOS uses a double-convex lens and optical bundle pipes to maintain high light intensity and signal data are obtained at 1000 Hz. In addition, its sampling flow rate (0.9 L min$^{-1}$) and optical unit temperature were actively controlled. The measurement interval was set to 1 min, which was then averaged to 1 h for further analysis. The default mass absorption cross section (MAC) of 10 m$^2$ g$^{-1}$ was applied for the derivation of $m_{BC}$. The lowest detection limit of COSMOS at 1 min time resolution is 50 ng m$^{-3}$. On an hourly basis, COSMOS can measure $m_{BC}$ in the range of 1–3000 ng m$^{-3}$ with an average accuracy of ~10 %, as compared with measurements by a single particle soot photometer (SP2) (Moteki and Kondo, 2010); and its sensitivity to the changes in the BC size distributions was less than 10 %, within the typical BC sizes in ambient atmosphere (Ohata et al., 2019). The SP2 is used as a reference instrument in previous studies (e.g., Ohata et al., 2019; Sinha et al., 2017). Further details about the measurement principles of COSMOS can be found in previous studies (Ohata et al., 2019; Kondo et al., 2009).

The Aethalometer uses the absorption of light at a wavelength of 880 nm by ambient aerosols collected on a quartz filter tape to determine the BC concentration. The flow rate was set to 5 L min$^{-1}$ and the accumulation area of the filter is 1.67 cm$^2$. The filter was set to change every 24 hours to minimize the loading effects. The data integration time was set to 5 min, which was then averaged to 1 h for further analysis. The default manufacturer-provided MAC value of 16.6 m$^2$ g$^{-1}$ was applied. The lower detection limit of the Aethalometer at 1 h time resolution is 20 ng m$^{-3}$ with the error limit of $\pm$30 ng m$^{-3}$. Comparison between $m_{eBC}$ and $m_{BC}$ (COSMOS) for cruises in 2017 and 2018, when both data are available, shows that the two data are in high consistency (Pearson correlation coefficient $R$>0.96) and that $m_{eBC}$ was 1.3–2.5 times $m_{BC}$ (COSMOS) (Fig. S2). Previous studies also show that the default parameter settings of the Aethalometer, as mentioned above, may cause the obtained BC mass concentrations to be 1–3 times the mass measured by SP2, depending on the sources and mixing states of the BC aerosols (Wang et al., 2014; Sharma et al., 2017; Laing et al., 2020). Due to the above reasons, the AE22 data in this study are mainly used as a reference. Hereinafter, for cruises conducted from 2016 to 2019, the analysis primarily relied on COSMOS data. In the case of the 2020 cruise, when only AE22 data was available, AE22 data was utilized for the analysis." (P4, L13 - P5, L21)

Regarding the BC correction for the Arctic region as proposed by Backmann et al., 2017, we would rather use the standard BC concentration recommended by the manufacturer at this stage, since the assumptions for the specific AE22 BC data correction may cause a wider error limit because the measurements covered a quite wide range of latitudes.

*If I understood correctly, the AE33 was used in 2020, otherwise the COSMOS was deployed. The wavelength dependence of the AE33 measurement is also often used to differentiate between biomass burning and anthropogenic BC. At least for 2020, you could do this differentiation and compare that to the model results. That would be a nice validation for the model, considering the moderate agreement between the measured and modeled total BC concentration.*

Reply> The model of the Aethalometer used in this study is AE22, not AE33. This typo error has been corrected in the revised manuscript. Since the AE22 model has only 2 channels, we decided not to apply source apportionment analysis using AE22 BC data.

*Specific comments:*

1. *P2,L9: "can work as CCN" please rephrase, e.g. to can act as*

   Reply> The word "work" has been replaced with "act". (P2, L10)

2. *P4, L8: Please mention already here, that the specific placement of your inlet was not the only method you have used to get rid of the ship emission influence, and that that the data sorting will be explained later.*

   Reply> The original expression "The air intake was set at the handrail of the front upper deck to avoid ship exhaust pollution." has been modified to "The air intake was set at the handrail of the front upper deck to prevent contamination from ship exhaust pollution. Furthermore, detailed information regarding data filtering techniques to mitigate the impact of ship exhaust will be provided later.". (P4, L9-11)

3. *P4, L10-22: what mass absorption efficiency is used in the COSMOS instrument?*

   Reply> The mass absorption cross-section used in COSMOS is 10 $m^2$ $g^{-1}$. This has been included in the revised manuscript. (P4, L31)

4. *P4, L23: you state that the BC mass concentration is derived from the AE33 880 nm measurements. What about the other wavelengths? Why not another one? The COSMOS uses 565nm, why not the closest wavelength of the AE33 was used to be more consistent?*

   Reply> The model of the Aethalometer used in this study is AE22, not AE33. This typo error has been corrected in the revised manuscript. The other AE22 channel uses 370 nm, of which values are interfered with by contributions from organic materials and are not suitable for direct comparison with the COCMOS 565 nm BC data.

5. *P4, L28-29: did you use these "default parameter settings"? And please provide more details about this fact. Please see my general comment about requesting much more details on how BC concentration was derived, and what are your uncertainties!*

Reply> The corrected and modified text on **pages 4 and 5** of the revised manuscript contains the mentioned information. Please see our response to the general comments.

6. *P5, L13-22: If I understood correctly, your method of discriminating the data which might be influenced by the ship emissions was the following: firstly the discrimination by wind direction, and in the years when the mixing ration of O3 was also measured you used that as an extra criterion. How much more data was considered invalid using the O3 criteria in 2017 and 2018? Which was not already discriminated by the wind direction? Can you estimate how much bias could it cause not having O3 data for the other years, and with that still using some data that might have been influenced by the ship?*

Reply> Using the $O_3$ criteria, 100 and 173 1-min $m_{BC}$ (COSMOS) data were considered invalid in 2017 and 2018, respectively. They accounted for less than 0.3 % and 0.4 % of the total amounts of valid data in 2017 and 2018, respectively. That is, further scrutiny by $O_3$ data had little influence on the overall properties of the observed BC. This point has been included in the manuscript as:

"It is noteworthy that the additional scrutiny based on the $O_3$ criteria had minimal impact on the overall characteristics of the observed BC by COSMOS. This screening process resulted in the exclusion of less than 0.3 % and 0.4 % of the total valid data in 2017 and 2018, respectively." (P6, L12-14)

7. *Figure 2, panel a: could you make the y axis logarithmic? Or do something else to see the lower values as well?*

Reply> To enhance the visibility of the lower values in Fig. 2a, a zoomed-in view of Fig. 2a with the y-axis maximum set to 80 ng m$^{-3}$ is shown in Fig. S3b of the revised supplementary.

8. *Figure 3: a logarithmic y axis for the BC concentration would help here a lot too. Caption: "short dashed gray line" it is enough if you write dashed gray line, I was looking for short lines that are dashed and gray :).*

Reply> The y-axis of Fig. 3 has been modified to a logarithmic scale. The word "short" has been removed from the caption. (P15, L3)

9. *P6, L30-31: This is what you do not see based on Fig2a, and therefore would be a logarithmic axis nice*

Reply> To enhance the visibility of the lower values in Fig. 2a, a zoomed-in view of Fig. 2a with the y-axis maximum set to 80 ng m$^{-3}$ is shown in Fig. S3b of the revised supplementary.

10. *P7, L25-26: what was this similar value reported by Taketani et al.?*

Reply> After careful examination of the data provided by Taketani, we found that the mean $m_{BC}$ reported by Taketani et al. (2022) for the same period is 0.6 (±0.8) ng m$^{-3}$, much lower than the value of 8.2 (±6.0) ng m$^{-3}$ obtained in our study. Consequently, we have elected to remove this section from the revised manuscript. We apologize for any confusion this may have caused.

11. *P11, L13: "anthropogenic productive activities producing" strange wording, please change*

Reply> The expression "anthropogenic productive activities producing" has been modified to "industry activities producing". (P15, L16)

12. *P11, L14: "may export" please change to "may be exported"*

Reply> The expression "may export" has been changed to "may be transported". (P15, L17)

13. *P11, L26-27: if I understood correctly, the AE33 and the COSMOS were not running parallel, and the AE33 measured in 2020 and otherwise the COSMOS. You state that the background periods were defined based on the measurements of the COSMOS. Does this mean that you did not even try to look for background periods in 2020? Why?*

Reply> Thank you for the comments. First, the model of the Aethalometer used in this study is AE22, not AE33. This typo error has been corrected in the revised manuscript. Second, the definition of background periods has been modified in the revised manuscript. The new definition includes background periods from 2016 to 2020. For details on the new definition, see our response to the next comment.

14. *P11, L29: your definition of a background period included that the BC mass concentration had to be above 1ng/m3, which is the 1 h detection limit of the instrument. My problem is with this, that with such a criterion you define that your background BC mass concentration has to be above 1 ng/m3. I understand that you are not able to reliably measure such low concentrations, but these values should be considered, otherwise the background concentration that you report is false, and at most you can call it as the upper limit, and the real background concentration could still be well below this value.*

Reply> The definition of the background periods has been modified from "The background periods in the western central Arctic Ocean (>72° N) were defined according to the $m_{BC}$ measured by COSMOS and the 5 day HYSPLIT back trajectories as follows: the 1-min $m_{BC}$ was below the lowest detection limit of 50 ng m$^{-3}$ for continually 2 hours or longer, the 1-h $m_{BC}$ was above the lowest detection limit of 1 ng m$^{-3}$, and the air masses were from the Arctic Ocean." to

"The background periods in the western central Arctic Ocean (>72° N) were determined according to the following criteria: first, for each hour with effective BC data, all three 5-day HYSPLIT back trajectories initiated at starting heights of 10, 500, and 1000 m originated from the Arctic Ocean. Additionally, all 1- min $m_{BC}$ or 5-min $m_{eBC}$ data within that hour were not removed due to ship exhaust according to data screening criteria described in Sect. 2. The second criterion is to ensure the accuracy of the selected data." (P15, L30-34)

15. *P11, L33- P12, L2: You state that the background value (that you have determined) being smaller that the overall mean BC concentration measured during a full year in the central Arctic suggests that even in the summer months the central Arctic is influenced by imported BC pollutants. Please compare the background to the mean summer values that were measured during MOSAIC. And if the relation is still the same, then your statement will be correct.*

Reply> Thank you for the comments. Following the revision of the selection criteria of background periods, this part has been removed from the revised manuscript.

16. *P12, L6-7: "The result should be representative as the data covered the summer seasons of three years." Why? Representative for what? I would just delete this statement, or make it more specific, please.*

Reply> Thank you for the comment. This statement has been deleted from the revised manuscript. We apologize for any confusion this may have caused.

17. *Section 4.4: you use the GEOS-Chem simulated biomass burning BC to total BC ratio to evaluate the high BC episodes as well. Table 2 shows in some cases of the episodes, enormous difference between the measured and modelled BC concentration (e.g. E1 20 vs 1 ng/m3) exits. In such cases, if the modelled vs. the measured BC concentration are many factors away from each other, I do not think that you can use the modelled biomass burning to total BC ratio to assume anything about the real measurement. Even for 2 out of the 3 selected episodes, the modelled BC mass is quite far away from the measured. Please comment on this.*

Reply> Despite significant differences between measured and modeled BC concentrations in certain episodes, we contend that using GEOS-Chem model simulation results to estimate the biomass burning to total BC ratio during the selected episode periods is justified for two main reasons. Firstly, the model, capable of reproducing 44% of the temporal spatial variation of the overall observed BC (Sect. 4.3), indicates the ubiquitous dominance of biomass burning BC North of 65 °N, where all episodes were identified (Fig. 3). Secondly, uncertainty analysis, involving shifting the episode period back or forward by 18 hours while maintaining its length, revealed changes of no more than 10 % in the modeled biomass burning to total BC ratio for most episodes, except for E2 and E6, as illustrated in the figure below.

[Figure]

**Figure R1-1**: The biomass burning to total BC ratio for each episode (line and circle marker) estimated from the GEOS-Chem model simulation results. Also shown in the figure are the ratios derived by moving each episode period back (square marker) or forward (triangle marker) by 18 hours.

This point has been included in the revised manuscript as: "Note that despite substantial normalized mean biases in model simulations compared to observed $m_{BC}$ for these episodes, ranging from −95 % to 178 %, we consider it reasonable to estimate the contribution of biomass burning to the total BC based on the model results. This is attributed to the pervasive dominance of biomass burning BC north of 65 °N, where all episodes were identified (Fig. 3). The estimate is further supported by the uncertainty analysis, involving shifting the episode period back or forward by 18 hours while maintaining its length, which revealed changes of no more than 10 % in the modeled biomass burning to total BC ratio for most episodes." (P18, L8-14)

Moreover, E3, E8, and E10 were selected for source and transport analyses because their temporal and spatial variations were well reproduced by the GEOS-Chem model. The analysis results were supported by the HYSPLIT back trajectory model. This point is better clarified in the revised

manuscript by modifying the original expression "Furthermore, the temporal and spatial variations of E3, E8, and E10 were well reproduced by GEOS-Chem model (Fig. 3). Therefore, in the following, the sources of BC during E3, E8, and E10 are elaborated based on GEOS-Chem model as well as HYSPLIT back trajectory model." to

"Additionally, the temporal and spatial variations of E3, E8, and E10 were well reproduced by GEOS-Chem model, showing nearly simultaneous peaks in observed and model data during these episodes (Fig. 3). Therefore, in the following sections, the sources and transports of BC during E3, E8, and E10 are elaborated based on GEOS-Chem model, with findings further corroborated by the HYSPLIT back trajectory model.". (P18, L14-18)

18. *P13, L11-15: Question regarding to the definition of the high BC episodes? Did you also use a geographic restriction? Only taking periods when the vessel's latitude was higher than a certain value? Because looking at Figure 3 it looks like that. I do see periods, that look long enough and have high BC concentrations but were not considered as an episode at low latitudes.*

Reply> Yes, we do restrict the episode selection to being north of 65° N. Therefore, the original expression "To characterize the sources of the high concentrations of BC in the Arctic Ocean and the marginal seas," has been modified to "To characterize the sources of the high concentrations of BC in the Arctic Ocean and the marginal seas (north of 65° N),". (P17, L29-31)

19. *P13, L13-14 and Table2: you state that the definition of the episodes included that the mean of the 1h BC concentration has to be above 20 ng/m3. How can it be that the overall observed mean BC is below 20 ng/m3 for E1 (first row of Table 2)?*

Reply> We apologize for the mistake. We have modified the expression "the mean of valid 1-h $m_{BC}$ during the defined periods was greater than 20 ng m$^{-3}$." to "the mean of valid 1-h $m_{BC}$ during the defined periods was not less than 20 ng m$^{-3}$.". (P17, L32-33)

In addition, we have rounded all $m_{BC}$ values not less than 10 ng m$^{-3}$, including those in Table 2, to integers.

20. *Figure 4: Please change the color for the ship positions, it is very hard to see it. And the wind arrows are also hard to see. (also for all other similar figures)*

Reply> Figure 4 and other similar figures have been modified to clearly show ship positions and wind arrows.

21. *Figure S5: You cannot see the BCbb sources, the dots are so small. Anyway: not only for S5 but also for Figure 4, S6, why don't you show a zoomed in picture of the region which you are talking about, one could see everything much better on the plots.*

Reply> Figures S9 (S5 in the preprint), S10 (S6 in the preprint), and 4 have been modified to show a zoomed in view of the region related to Episode 3. Besides, Figure S9 as well as Figures S11 and S14 (S7 and S10 in the preprint) are modified to clearly show the BCbb sources by coarsening the longitudinal and latitudinal dimensions by a factor of 5.

22. *Figure 5: same as for Figure 4, actually do it please for all similar figures.*

Reply> Figures 4 and 5 and other similar figures have been modified to clearly show ship positions and wind arrows.

23. *P16, L20: "CO2 and O3 were not or even slightly decreased" I see there quite a decrease. Maybe use other wording.*

Reply> When compared with CO and CH$_4$, the decreases in CO$_2$ and O$_3$ do not show significant differences from their temporal changes before and after episode 8. Therefore, we will retain the original statement. (P21, L21)

24. *P16, L23-27: I only see a decrease of CO2 when the bb plume was present. All the possible explanations would rather explain a constantly low CO2 concentration, am I not right?*

Reply> While we agree that CO$_2$ uptake by high latitude vegetation may explain the constantly low observed CO$_2$ concentrations, we still think that this uptake might have contributed to the slight decrease in CO$_2$ concentrations observed at a location distant from the continent. This consideration is in addition to acknowledging the possible influence of smoldering combustion conditions on that minor decrease.

Accordingly, we retain the original statement, but modify "The decrease" to "The slight decrease". (P21, L24)

25. *P17, L5-6: why was the background BC not subtracted from mBC/ΔCO?*

Reply> To ensure that there were sufficient data to characterize the spatiotemporal changes in $m_{BC}/\Delta CO$, the background of $m_{BC}$ was not subtracted. Actually, after subtracting the background $m_{BC}$ of 2.1 ng m$^{-3}$ (Table 1), 53 % of the $m_{BC}/\Delta CO$ data are lost.

Therefore, the original expression "note that in order to obtain more $m_{BC}/\Delta CO$ data points, the background of $m_{BC}$ was not subtracted" has been modified to "note that in order to ensure that there were sufficient data to characterize the spatiotemporal changes in $m_{BC}/\Delta CO$, the background of $m_{BC}$ was not subtracted". (P22, L6-8)

*26. P23, L2: "at higher than low latitude regions" ???*

Reply> It has been modified to "at higher latitude regions". (P29, L5)

---

## Author Comment (AC4)

**Response to Anonymous Referee #2**

We appreciate the invaluable comments. Our answers to the comments are provided below. The reviewer comments are written in italics.

*Review of the manuscript titled 'Measurement report: Shipborne observations of black carbon aerosols in the western Arctic Ocean during summer and autumn 2016–2020: boreal fire impacts'*

*The manuscript titled 'Shipborne Observations of Black Carbon Aerosols in the Western Arctic Ocean During Summer and Autumn 2016–2020: Boreal Fire Impacts' authored by Deng et al., presents a comprehensive analysis of black carbon levels in the Arctic Ocean during the summer and autumn seasons. Furthermore, it conducts an in-depth examination of the emission sources contributing to elevated black carbon episodes using simulations from a global chemistry-transport model. This manuscript contributes a significant and valuable dataset to the Arctic region, which is often limited in available data. After addressing the major comments outlined, I highly recommend this manuscript for publication in ACP (Atmospheric Chemistry and Physics).*

*The manuscript details the analysis of 10 specific episodes of black carbon (BC), yet it overlooks the explanation of additional episodes present in the dataset. There's a notable lack of comprehensive explanations for the identification of these episodes and the criteria utilized in the selection process for these events. Additionally, the manuscript employs a model with a notably coarse resolution for simulations, which might introduce biases into the results, particularly when applied to Arctic conditions. The authors must provide a more detailed account of the uncertainties inherent in the model and highlight potential biases when interpreting the simulated values within the manuscript. Addressing these concerns will significantly strengthen the manuscript's scientific rigor and ensure a more robust interpretation of the findings.*

Reply> We appreciate the reviewer's valuable comments. We apologize for the incomplete explanation of the selection process for high BC episodes in the preprint. In the preprint, the criterion of 10 ng m$^{-3}$ represents three times the background $m_{BC}$ level determined in Sect. 4.2 of the preprint. Furthermore, considering that longer episodes and higher BC mass concentrations would offer better representation, we refined the selection criteria to ultimately identify approximately 10 episodes. In the revised manuscript, although the estimated background $m_{BC}$ level changed due to the revision of the background period definition, we retained this definition but modified the presentation.

The original expression regarding the definition of high BC episodes "To characterize the sources of the high concentrations of BC in the Arctic Ocean and the marginal seas, high BC episodes were defined as periods when the 1-h $m_{BC}$ was continually greater than 10 ng m$^{-3}$ for 18 h or longer and the mean of valid 1-h $m_{BC}$ during the defined periods was greater than 20 ng m$^{-3}$. In total, 10 high BC mass concentration episodes were identified (Figs. 1, 3, S1, and S3, Table 2)." has been modified to

"To characterize the sources of the high concentrations of BC in the Arctic Ocean and the marginal seas (north of 65° N), we identified periods when the 1-h $m_{BC}$ exceeded 10 ng m$^{-3}$. From these periods, we further selected those lasting 18 h or longer and the mean of valid 1-h $m_{BC}$ during the selected period was not less than 20 ng m$^{-3}$. This process allowed us to identify and refine 10 high BC episodes (Figs. 1, 3, S1, and S6, Table 3).". (P17, L29-P18, L1)

As for the other high BC episodes than E3, E8, and E10 that have been explained in detail in the manuscript, their temporal and spatial variation failed to be well reproduced by GEOS-Chem model. From this point of view, it is inappropriate to interpret their origins based on the model result. However, due to the ubiquitous dominance of biomass burning BC in the Arctic Ocean region where those episodes occurred, the biomass burning to total BC ratio was estimated based on the model results in the manuscript. This estimate is further supported by the uncertainty analysis, involving shifting the episode period back or forward by 18 hours while maintaining its length, which revealed changes of no more than 10 % in the modeled biomass burning to total BC ratio for most episodes, except for E2 and E6. This is illustrated in the figure below.

[Figure]

This point raised in the preprint is supplemented in the revised manuscript as follows: "Note that despite substantial normalized mean biases in model simulations compared to observed $m_{BC}$ for these episodes, ranging from −95 % to 178 %, we consider it reasonable to estimate the contribution of biomass burning to the total BC based on the model results. This is attributed to the pervasive dominance of biomass burning BC north of 65 °N, where all episodes were identified (Fig. 3). The estimate is further supported by the uncertainty analysis, involving shifting the episode period back or forward by 18 hours while maintaining

its length, which revealed changes of no more than 10 % in the modeled biomass burning to total BC ratio for most episodes." (P18, L8-14)

Although coarse horizontal resolution may cause larger uncertainty in model simulation results, other mechanisms including transport, deposition, and emissions may play more important roles in determining the uncertainty range of the model. In fact, a multi-model study by Whaley et al. (2022) indicates that models with higher resolution didn't produce more accurate simulations of BC mass. Therefore, we will not re-run the model with a higher resolution.

Uncertainty analysis of the simulations is presented in Sect. 4.3 of the revised manuscript.

*Line 20 (Page-1): I would suggest specifying the particular seasons under consideration here. Failing to do so might lead to a misconception that the findings represent an annual average. Clarifying the specific seasons, such as summer and autumn, within the analysis would prevent any misinterpretation regarding the temporal scope of the data and offer a more accurate representation of the findings.*

Reply> The expression "in 2019" has been modified to "by the 2019 cruise". And the whole sentence has been modified from "The average $m_{BC}$ in the surface air over the Arctic Ocean (72–80° N) observed in 2019 was over 70 ng m$^{-3}$, which was substantially higher than in other years (approximately 10 ng m$^{-3}$)." to "The average $m_{BC}$ in the surface air over the Arctic Ocean (72–80° N) observed by the 2019 cruise exceeded 70 ng m$^{-3}$, which was substantially higher than cruises in other years (approximately 10 ng m$^{-3}$).".(P1, L19-21)

*Line 27 (Page-1): what does this mean " with some near-surface and others in the mid-troposphere"?*

Reply> The original sentence "The model analysis indicated that most episodes were attributed to the airmasses transported from boreal fires to the Arctic Ocean, with some near-surface and others in the mid-troposphere." has been modified to "The model analysis indicated that most episodes were attributed to BC containing airmasses transported from boreal fire regions to the Arctic Ocean, with some transport occurring near-surface and others in the mid-troposphere.". (P1, L26-28)

*Line 28-30 (Page-1): The abstract appears to contain repetitive sentences. I suggest that the authors revise the abstract to convey essential information more concisely. Condensing the content while retaining crucial details will enhance the abstract's clarity and effectiveness in summarizing the study's key findings and contributions.*

Reply> Thank you for your valuable comment. Upon review, we have maintained the majority of the abstract content and made modest alterations. (Page 1)

*Line 2 (Page-2): AMAP, 2021b ? Where is 'a'?*

Reply> The reference "AMAP, 2021b" has been changed "AMAP, 2021a". (P2, L4)

*Line 7 (Page-2): can include? Rephrase the sentence.*

Reply> The word "can" has been changed to "may". (P2, L7)

*Line 13(Page-2): Add relevant citation*

Reply> A citation to "AMAP, 2011: The Impact of Black Carbon on Arctic Climate." has been added to the revised manuscript. (P2, L13)

*Line 18 (Page-2): I would recommend citing Gogoi et al., 2021 (Long-term changes in aerosol radiative properties over Ny-Ålesund: Results from Indian scientific expeditions to the Arctic) for the continuous long term monitoring of aerosols in the Arctic. Further, I would recommend comparing the values in this study with Gogoi et al., 2021.*

Reply> Thank you for the suggestion. In addition to the cited works and the Gogoi et al. (2021) recommended by the reviewer, there are other papers describing continuous long-term monitoring of BC aerosols in the Arctic that are worthy of being acknowledged. Therefore, instead of including Gogoi et al. (2021) into the reference, we have modified the expression from "(Stohl et al., 2013; Schmale et al., 2022)" to "(e.g., Stohl et al., 2013; Schmale et al., 2022)" to provide examples. (P2, L18-19)

Moreover, due to the significant spatial separation between our study's measurements and those reported by Gogoi et al. (2021), we have chosen not to directly compare the values obtained in our study with theirs.

*Line 32 (Page-2): What do authors want to convey here? It is obvious that the BC concentration reduces when the observed location is far from the source region.*

Reply> The phrase "source region" has been modified to "continent". In addition, the whole sentence "They also revealed important characteristics of the spatial distribution of BC in the Arctic Ocean and its marginal seas such as the BC concentration decreases with the growing distance from the source region (Xie et al., 2007; Sakerin et al., 2015, 2021)." has been modified to "They also revealed important characteristics of the spatial distribution of BC in the Arctic Ocean, demonstrating that BC concentration diminishes in the northern direction and decreases as distance from the continent increases (Xie et al., 2007; Sakerin et al., 2015, 2021).". (P2, L31-P3, L1).

*Line 29-32 (Page-3): Rewrite the sentence.*

Reply> The original sentence "BC monitors based on light absorption theory were operated during the 2016–2020 summer and autumn expeditions from the North Pacific Ocean to the Arctic Ocean, encompassing the western Arctic Ocean and part of the east Siberian Sea, and back to the North Pacific Ocean to measure the BC mass concentration ($m_{BC}$)." has been modified to "to enhance comprehension of the distribution and sources of BC in the Arctic, the mass concentration of BC ($m_{BC}$) was monitored across five round-trip expeditions conducted between the North Pacific Ocean and the Arctic Ocean during the summer and early autumn of 2016–2020." (P3, L30-32)

*Line 28 (Page 4): What does the 'default parameter settings' mean here? Please explain. Also, explain more about the loading effect corrections and other corrections applied to the data. I would also recommend the authors to explain the uncertainties in the measurements. COSMOS measures at 565 nm, and Aethalometer measurements were at 880 nm. How did the authors compare these measurements at two distinct wavelengths?*

Reply> The model of the Aethalometer which was deployed during the cruise is AE22, not AE33. This mistake has been corrected in the revised manuscript. Details including the default parameter settings, uncertainties, and a comparison between the two instruments have been added to **paragraphs 2-4, Sect. 2** of the revised manuscript.

As for details on loading effect corrections and other corrections for the measurements, they are beyond the scope of this paper and will not be included.

*Line 13 (Page 5): To avoid the influence of ship exhausts, the authors have used 1- or 5-min data records that occurred when the 1-min wind direction and speed relative to the ship's course within ±60° of the bow and >3 m s−1. Why is it done so for these specific values? What are the reasons for choosing these specific limits?*

Reply> We determined the data screening criteria by experimenting with various criteria as well as referring to previous studies, such as Taketani et al. (2016). Other criteria we tested include requiring the 1-min wind direction and speed relative to the ship's course to be within ±70° of the bow and >2 m s$^{-1}$, respectively. Ultimately, we selected wind direction within ±60° of the bow and a wind speed >3 m s$^{-1}$. This choice effectively screened out data associated with simultaneous significant decreases in $O_3$ and increases in BC observed in the open North Pacific Ocean (approximately north of 42° N) and the Arctic Ocean.

Regarding the BC mass concentrations obtained south of 42° N during the 2017 and 2018 cruises, we subjected them to additional scrutiny. Whenever there was a simultaneous decrease in $O_3$ and an increase in BC, we considered both 1-minute BC and $O_3$ data as invalid. This additional scrutiny resulted in the

exclusion of only 0.12 % and 0.17 % of the observed data from the 2017 and 2018 cruises, respectively. This information has been added to the revised manuscript as:

"It is noteworthy that the additional scrutiny based on the $O_3$ criteria had minimal impact on the overall characteristics of the observed BC by COSMOS. This screening process resulted in the exclusion of less than 0.3 % and 0.4 % of the total valid data in 2017 and 2018, respectively.". (P6, L12-14)

***Line 26 (Page 5): Since the horizontal resolution of GEOS-chem was 2° × 2.5°, how is it valid over the Arctic regions due to the higher grid resolution?***

Reply> Although coarse horizontal resolution may cause larger uncertainty in model simulation results, other mechanisms including transport, deposition, and emissions may play more important roles in determining the uncertainty range of the model. In fact, a multi-model study by Whaley et al. (2022) indicates that models with higher resolution didn't produce more accurate simulations of BC mass. Therefore, we will not re-run the model with a higher resolution.

***Line 25 (Page 7): Authors have compared the measurements of BC using an Aethalometer and SP2 here. I would not recommend the direct comparison of the measurements with SP2 since the values are highly dependent on the BC size distribution. SP2 has lower and upper bounding limits for the measured BC distributions. This will result in a biased comparison.***

Reply> First, the comparison was between COSMOS (not Aethalometer) and SP2. According to Ohata et al., 2019 (Accuracy of black carbon measurements by a filter-based absorption photometer with a heated inlet), the COSMOS can measure $m_{BC}$ in the range 1–3000 ng m$^{-3}$ with ~10 % accuracy as compared with SP2. Furthermore, despite the size-dependence observed in the comparison between COSMOS and SP2, the estimated sensitivity of the COSMOS $m_{BC}$ values, with a 1 h time resolution, to changes in the BC size distributions was less than 10%, falling within the typical natural variabilities of BC size distributions. Therefore, we will maintain our comparison between COSMOS and SP2. Please note that the accuracy of the comparison between COSMOS and SP2 measurements has been addressed in the preprint. In the revised manuscript, this point is addressed even clearer as follows:

"On an hourly basis, COSMOS can measure $m_{BC}$ in the range of 1–3000 ng m$^{-3}$ with an average accuracy of ~10 %, as compared with measurements by a single particle soot photometer (SP2) (Moteki and Kondo, 2010); and its sensitivity to the changes in the BC size distributions was less than 10 %, within the typical BC sizes in ambient atmosphere (Ohata et al., 2019). The SP2 is used as a reference instrument in previous studies (e.g., Ohata et al., 2019; Sinha et al., 2017)." (P5, L1-5)

*Figure 1: I would recommend modifying the color bar to a different color option (either jet or something else). It is difficult to identify the variability using these colors.*

Reply> The color table has been revised to WarmCold.

*Line 15 (Page 9): I would recommend modifying the plots with Lower whisker – 1st percentile, upper whisker – 99th percentile.*

Reply> Due to the considerably high values of the 99th percentile of the BC data, we have chosen to retain the 9th percentile as the lower whisker and the 91st percentile as the upper whisker. Additionally, we have included a supplementary figure (Fig. S3) that presents the entire range of observed data, along with a focused view specifically showcasing data within the range of not exceeding 80 ng m$^{-3}$.

*Figure 3: I would recommend changing the left y-axis to a logarithmic scale*

Reply> The left y-axis of Fig. 3 has been modified to a logarithmic scale.

*Line 6 (Page 11): The derived background concentrations in this study are misleading. The criteria used here need to be proven.*

Reply> The definition of background periods have been modified from "The background periods in the western central Arctic Ocean (>72° N) were defined according to the $m_{BC}$ measured by COSMOS and the 5 day HYSPLIT back trajectories as follows: the 1-min $m_{BC}$ was below the lowest detection limit of 50 ng m$^{-3}$ for continually 2 hours or longer, the 1-h $m_{BC}$ was above the lowest detection limit of 1 ng m$^{-3}$, and the air masses were from the Arctic Ocean." to

"The background periods in the western central Arctic Ocean (>72° N) were determined according to the following criteria: first, for each hour with effective BC data, all three 5-day HYSPLIT back trajectories initiated at starting heights of 10, 500, and 1000 m originated from the Arctic Ocean. Additionally, all 1-min $m_{BC}$ or 5-min $m_{eBC}$ data within that hour were not removed due to ship exhaust according to data screening criteria described in Sect. 2. The second criterion is to ensure the accuracy of the selected data." (P15, L30-34)

*Line 13 (Page 11): Rewrite the sentence.*

Reply> The original expression "Those activities include anthropogenic productive activities producing large amount of air pollutants in low latitude regions that may export to the Arctic through long-range transport (Ikeda et al., 2017; Zhu et al., 2020)," has been modified to "Those activities include industry activities producing large amounts of air pollutants in lower latitude regions that may be transported to the Arctic through long-range transport (Ikeda et al., 2017; Zhu et al., 2020),". (P15, L15-17)

*Line 27 (Page 11): Why the authors have chosen 5 days airmass trajectory for this study? I would suggest to explain the reason here.*

Reply> The 5-day duration was chosen because this time is long enough to be able to indicate the possible source regions of high BC episodes as well as short enough to ensure the accuracy of the trajectories. This is mentioned in the revised manuscript: "The selection of a 5-day duration allows for identifying potential source regions of high BC episodes (Sect. 4.4) while ensuring trajectory accuracy (Backman et al., 2021)." (P7, L5-7)

*Line 10 (Page 12): I would suggest that the authors compile the various comparisons made regarding black carbon concentrations observed in their study with those from other relevant studies in the Arctic region. This compilation could be organized into a table format detailing concentrations, seasons, instruments used, and corresponding references, etc. This tabulated presentation would provide a comprehensive and accessible comparison, aiding readers in understanding the context and variability of BC concentrations in the Arctic across different studies.*

Reply> Thank you for the valuable comments. In the revised manuscript, we have incorporated a table (Table 2) containing BC mass concentrations derived from shipborne observations reported in previous studies. Based on this information, we have revised the comparison between this study and previous studies from:

[revised manuscript text omitted]

***Line 17 (Page 12): I would recommend the authors show R2 values than R throughout the manuscript.***

Reply> We present the Pearson correlation coefficient, denoted as *R*. Hence, we will consistently use *R* instead of showing $R^2$.

*Table 2: I would recommend showing the region for these episodes in this table.*

Reply> The regions for these episodes have been compiled as longitudinal and latitudinal ranges in Table 3.

*Line 10 (Page 15): I would recommend the authors to compare the median values of BC for all these episodes since the mean will not represent the actual variabilities.*

Reply> The median and mean values for the 10 selected episodes are displayed in Table R2-1. With the exception of Episode 8, the differences between the median and mean values for each episode are not significant. Moreover, we think that the mean values better represent the magnitude of the increment of each episode from the background. Consequently, we will continue to utilize mean values in the manuscript to characterize these episodes.

**Table R2-1**: The median and mean values (unit: ng m$^{-3}$) for the 10 selected episodes.

| Episodes | E1 | E2 | E3 | E4 | E5 | E6 | E7 | E8 | E9 | E10 |
|---|---|---|---|---|---|---|---|---|---|---|
| **Median** | 21 | 33 | 38 | 20 | 25 | 23 | 22 | 23 | 24 | 28 |
| **Mean** | 20 | 34 | 44 | 25 | 25 | 26 | 32 | 55 | 25 | 29 |

*Figure 6: I would recommend changing the y-axis to a logarithmic scale here for the BC mass concentration.*

Reply> The y-axis for the BC mass concentration has been modified to a logarithmic scale.

*Figure S1: I recommend changing the color options for the color bar. It is difficult to understand.*

Reply> The color table has been revised to WarmCold.

*Figure S2: I would recommend changing the axis to a logarithmic scale.*

Reply> Changing the axis of a box plot figure to a logarithmic scale proves to be challenging. As an alternative, we include a zoomed-in view of the figure as Fig. S5 in the revised supplementary.

*Figure S5, S7, and S10: These figure needs to be revised with detailed color contrasts.*

Reply> The color table of these figures has been updated. These figures are Figs. S9, S11, and S14 in the revised supplementary.

*Figure S14: Recommend to modify the color bar*

Reply> The color table has been revised to WarmCold. This figure is Fig. S18 in the revised supplementary.

*Figure S15: I would recommend showing all the episodes in this plot rather than removing the other episodes.*

Reply> Back trajectories of all the episodes are included in the revised figure, labeled as Fig. S19 in the revised supplementary.

---

## Author Comment (AC5)

**Response to Anonymous Referee #3**

We appreciate the invaluable comments. Our answers to the comments are provided below. The reviewer comments are written in italics.

*Measurement report: Shipborne observations of black carbon aerosols in the western Arctic Ocean during summer and autumn 2016–2020: boreal fire impacts*

*Summary*

*Yange Deng[1] and co-authors present a really valuable data set of mass concentrations of eBC within the Arctic Ocean, a region with sparse data coverage. The authors outline through the use of in situ shipborne measurements and a global chemistry transport model that the proportion of eBC mass arising from biomass burning increases with the latitude at which observations are performed.*

*I recommend this paper after the following corrections are made.*

*Major comments:*

*Method section I suggest*

*A lack of any description of how HYSPLIT was utilised i.e. which settings were chosen, whether it was a single or ensemble run, the initialised altitude, why 5-days was chosen, whether or not air masses below the mixed-layer were selected for (assume not given the altitude plots), which metrological fields were utilised. Please add more information about how HYSPLIT was utilised.*

Reply> Single runs of HYSPLIT were calculated with a time step of 1 hour, a starting height of 500 m above model ground level, and a duration of 5 days. The 5-day duration was chosen because this time is long enough to be able to indicate the possible source regions of observed high BC episodes as well as short enough to ensure the accuracy of the trajectories. Air masses below the mixed layer were included because air masses from boreal fires may transport through the mixed layer to the ship position. The NCEP's GDAS data was utilized to provide meteorological fields. Furthermore, in the revised manuscript, back trajectories initiated at starting heights of 10 and 1000 m above model ground level were calculated to assist in the selection of Arctic Ocean background periods.

The original expression "Furthermore, the NOAA Air Resources Laboratory Hybrid Single-Particle Lagrangian Integrated Trajectory model (HYSPLIT; Stein et al., 2015) was also applied to assist the interpretation of the BC sources." has been modified to

"Furthermore, backward trajectories were generated using the NOAA Air Resources Laboratory Hybrid Single-Particle Lagrangian Integrated Trajectory model (HYSPLIT; Stein et al., 2015) to aid in interpreting the sources of the observed BC and identifying background periods in the Western Arctic Ocean. These trajectories were calculated with a 1-hour time step, initiated at the ship positions with starting heights of 10, 500, and 1000 m above model ground level, and extended for 5 days. The selection of a 5-day duration allows for identifying potential source regions of high BC episodes (Sect. 4.4) while ensuring trajectory accuracy (Backman et al., 2021). The meteorological data used for HYSPLIT was the NCEP's GDAS data, featuring a horizontal resolution of 1° × 1° and 24 pressure levels extending from the ground to 20 hPa in the vertical direction.". (P7, L1-9)

*Please include the amount of data which is removed when doing the cleaning of the data. Also, include the amount of data which is below the limit of detection. What proportion of the measured observations is left after the various cleaning criteria are applied to the data set e.g. detection limits, averaging thresholds, and ship contamination.*

Reply> We mainly cleaned data that may have been contaminated by ship exhaust. After the cleaning process, all other data, including values below the limit of detection, were presented as recorded (e.g., Figs. 1b-1f, 2a, 3, and 6a). Moreover, Fig. S6 illustrates the BC mass concentration before the removal of data influenced by ship exhausts. The quantity of data removed due to potential ship contamination was included in the original manuscript, corresponding to **Lines 10-12 on Page 6** of the revised manuscript.

Specific information regarding data below the limit of detection post the exclusion of potentially ship-contaminated data is provided in the current manuscript: "Within the hourly BC mass concentration data, 5–13 % of COSMOS data and 63–71 % of Aethalometer data fall below their respective detection limits.". (P6, L16-18)

*More justification is required to support why a particle MAC is applied to the absorption coefficient measurements as opposed to another i.e. a more site-specific MAC. The paper could simply report the absorption coefficient instead?*

Reply> The model of the Aethalometer which was deployed during the cruise is AE22, not AE33. This mistake has been corrected in the revised manuscript. The AE22 defines BC from 880 nm absorption with the manufacturer provided MAC, sigma, value of 16.6 $m^2$ $g^{-1}$. COSMOS also uses the default instrument MAC, value of 10 $m^2$ $g^{-1}$. Since the measurements were done over a quite wide latitudinal range, we decided to use manufacturer recommended standard MAC rather than site specific values. Paragraphs 2 to 4 of Sect. 2 in the revised manuscript gives more information about the two instruments including the MAC applied.

Furthermore, we insist to report $m_{BC}$ instead of aerosol absorption coefficient because the main instrument we used for the measurements is COSMOS with a heated inlet, whose accuracy in measuring $m_{BC}$ has been critically assessed by previous studies (Ohata et al., 2019; Sinha et al., 2017). For a detailed explanation, please see our response to the reviewer's first specific comment on pages 5 and 6.

*Results and discussion section*

*Elevated values:*

*The authors highlight 10 events but the underlying reason for why the events are highlighted in the first place is not so clear, this maybe due to a lack of stated research questions.*

*Also, the explanation for why the criteria was used to defined the episodes was not explained i.e. when the 1-h $m_{BC}$ was continually greater than 10 ng m−3 for 18 h or longer and the mean of valid 1-h $m_{BC}$ during the defined periods was greater than 20 ng m−3.*

Reply> We selected the 10 high BC episodes to characterize the sources of the high concentrations of BC observed during the cruises. This point has been explained in the original manuscript, corresponding to **Line 30 on Page 17** of the revised manuscript.

The criterion of 10 ng m$^{-3}$ represents three times the background $m_{BC}$ determined in Sect. 4.2 of the original manuscript. Furthermore, considering that longer episodes and higher BC mass concentrations would offer better representation, we refined the selection criteria to ultimately identify approximately 10 episodes. The original expression regarding the definition of high BC episodes "To characterize the sources of the high concentrations of BC in the Arctic Ocean and the marginal seas, high BC episodes were defined as periods when the 1-h $m_{BC}$ was continually greater than 10 ng m$^{-3}$ for 18 h or longer and the mean of valid 1-h $m_{BC}$ during the defined periods was greater than 20 ng m$^{-3}$. In total, 10 high BC mass concentration episodes were identified (Figs. 1, 3, S1, and S3, Table 2)." has been modified to

"To characterize the sources of the high concentrations of BC in the Arctic Ocean and the marginal seas (north of 65° N), we identified periods when the 1-h $m_{BC}$ exceeded 10 ng m$^{-3}$. From these periods, we further selected those lasting 18 h or longer and the mean of valid 1-h $m_{BC}$ during the selected period was not less than 20 ng m$^{-3}$. This process allowed us to identify and refine 10 high BC episodes (Figs. 1, 3, S1, and S6, Table 3).". (P17, L29-P18, L1)

*Was there also a latitudinal criterion imposed on the event selection to study 'Arctic Ocean and the marginal seas' as all events are above 65 degrees norther, and either in the 'North of 72° N group' or 'between 52 and 72° N' group.*

Reply> Yes, we specifically restrict the episode selection to locations north of 65° N. Therefore, the original expression "To characterize the sources of the high concentrations of BC in the Arctic Ocean and the marginal seas," has been modified to "To characterize the sources of the high concentrations of BC in the Arctic Ocean and the marginal seas (North of 65° N),". (P17, L29-31)

However, it is worth noting that the limitation of north of 65° N does not align with the grouping of latitudes used to illustrate the temporal and spatial variability of BC mass concentrations in Section 4.1. This information is included in the revised manuscript as: "Note that the grouping mentioned here does not comply with the latitudinal constraints (i.e., north of 65° N) used to select high BC episodes in Sect. 4.4.". (P7, L25-27)

*Background:*

*It is not clearly explained what the authors are trying to achieve by presenting 'background' values. Do they mean values that are unaffected by anthropogenic emissions? Also, the criteria has no justification and seems arbitrary.*

Reply> We apologize for the misleading in our previous definition of the background periods. The background in the context of this manuscript refers to the preindustrial atmospheric conditions. Whereas finding a situation identical to preindustrial atmospheric conditions is challenging, examining periods in the Arctic Ocean unaffected by regional transport could offer insights into the preindustrial background conditions. This assumes that the impact of natural terrestrial activities, such as wildfires, on BC in the preindustrial Arctic Ocean atmosphere was negligible. This information has been included in the manuscript as:

"While finding a situation entirely identical to the preindustrial atmosphere is challenging due to the pervasive influence of anthropogenic activities on even natural events like wildfires (McCarty et al., 2021), examining periods in the Arctic Ocean unaffected by regional transport could offer insights into the preindustrial atmospheric situations. This assumes that the impact of natural terrestrial activities, such as wildfires, on BC in the preindustrial Arctic Ocean atmosphere was likely negligible, recognizing the inherent uncertainties in making such historical assessments.". (P15, L9-14)

The definition of background periods has been modified from "The background periods in the western central Arctic Ocean (>72° N) were defined according to the $m_{BC}$ measured by COSMOS and the 5 day HYSPLIT back trajectories as follows: the 1-min $m_{BC}$ was below the lowest detection limit of 50 ng m$^{-3}$ for continually 2 hours or longer, the 1-h $m_{BC}$ was above the lowest detection limit of 1 ng m$^{-3}$, and the air masses were from the Arctic Ocean." to

"The background periods in the western central Arctic Ocean (>72° N) were determined according to the following criteria: first, for each hour with effective BC data, all three 5-day HYSPLIT back trajectories initiated at starting heights of 10, 500, and 1000 m originated from the Arctic Ocean. Additionally, all 1-min $m_{BC}$ or 5-min $m_{eBC}$ data within that hour were not removed due to ship exhaust according to data screening criteria described in Sect. 2. The second criterion is to ensure the accuracy of the selected data.". (P15, L30-34)

*Supplement:*

*The supplement is of course important to explain interesting but not essential results. In this specific work, the supplementary is 18 pages long with 15 figures, are all necessary.*

*Figures S5 & S7: can the mass concentration of eBC not be grouped based on the metres AGL of the collocated back trajectories.*

Reply> For Episode 3, corresponding to Fig. S9 in the revised supplementary, the average $m_{BC}$ associated with back trajectories originating above 1.5 km AGL (49 ng m$^{-3}$) is higher than the average $m_{BC}$ associated with back trajectories originating below 1.5 km AGL (39 ng m$^{-3}$). For Episode 8, corresponding to Fig. S11 in the revised supplementary, the $m_{BC}$ cannot be well grouped based on the meters AGL, possibly due to the limited number of valid data collected during Episode 8.

We have decided not to include the above information in the revised manuscript because we think it contributes little to the interpretation of $m_{BC}$ transport.

*Figure S15: There is a mention of the fact that backward air mass trajectories (back trajectories) are initialised at 500 m above the ship.*

Reply> More detailed information on the back trajectories has been included in the revised manuscript as mentioned in our reply to the reviewer's first major comment.

*Specific comments:*

*The title is a bit misleading. The manuscript present absorption data, not BC data or eBC data. Justify why you made this decision.*

Reply> This is because the main instrument we used for the measurements was COSMOS with a heated inlet, whose accuracy in measuring $m_{BC}$ has been critically assessed by previous studies (Ohata et al., 2019; Sinha et al., 2017). We apologize that we haven't stated this clearly in the preprint. We apologize also for missing any consistency check between measurements by the two different instruments. These two points are included in the revised manuscript as follows:

"Whereas both instruments use light absorption methods, COSMOS was equipped with a 400 °C heated inlet line. This feature effectively eliminated interference from volatile non-refractory aerosol chemical species internally mixed with BC, ensuring a high accuracy of $m_{BC}$ measurement. This aspect has been critically assessed in previous studies (Ohata et al., 2019; Sinha et al., 2017). Consequently, COSMOS measurements differ from traditional light absorption methods, where the mass concentration of BC is referred to as equivalent BC (eBC, Petzold et al., 2013). Therefore, instead of using eBC, the term BC can be used for COSMOS data in a general sense (Ohata et al., 2019)." (P4, L15-21)

"Comparison between $m_{eBC}$ and $m_{BC}$ (COSMOS) for cruises in 2017 and 2018, when both data are available, shows that the two data are in high consistency (Pearson correlation coefficient $R>0.96$) and that $m_{eBC}$ was 1.3–2.5 times $m_{BC}$ (COSMOS) (Fig. S2). Previous studies also show that the default parameter settings of the Aethalometer, as mentioned above, may cause the obtained BC mass concentrations to be 1–3 times the mass measured by SP2, depending on the sources and mixing states of the BC aerosols (Wang et al., 2014; Sharma et al., 2017; Laing et al., 2020). Due to the above reasons, the AE22 data in this study are mainly used as a reference. Hereinafter, for cruises conducted from 2016 to 2019, the analysis primarily relied on COSMOS data. In the case of the 2020 cruise, when only AE22 data was available, AE22 data was utilized for the analysis." (P5, L12-21)

*Title: 'impacts of boreal fires'*

Reply> The original expression "boreal fire impacts" has been modified to "impact of boreal fires".

*Abstract:*

*L3: observations of BC ? L14?*

Reply> The original expression "BC" has been modified to "BC aerosols". (P1, L14)

*L15-17: Say the actual number of research expeditions*

Reply> The total number of research expeditions is disclosed in the final paragraph of the "Introduction" section in the revised manuscript. (P3, L31) This information is not included in the abstract as it does not constitute a key element.

*L20: replace 'over' with 'greater than'*

Reply> The original expression "was over" has been changed to "exceeded". (P1, L20)

*L22: perhaps due to 'that' more frequent wildfires 'had occurred' in the Arctic… or perhaps due to more frequent wildfires 'occurring' …*

Reply> The word "occurred" has been changed to "occurring". (P1, L22)

**L23: biomass burning contributed the most to the observed BC in …**

Reply> The original expression "composed the largest contribution" has been modified to "contributed most". (P1, L23)

**L23: by mass**

Reply> The expression "to the observed BC in the" has been modified to "to the observed BC by mass in the". (P1, L23)

**L26: change to transported from regions with boreal fires**

Reply> The original expression "transported from boreal fires to the Arctic Ocean," has been modified to "transported from boreal fire regions to the Arctic Ocean,". (P1, L26-27)

**Introduction:**

**L2: 'climate warming rate' be more specific e.g. average surface temperature**

Reply> The original expression "The climate warming rate in the Arctic is more than three times of the global average," has been modified to "The annual average surface temperature increase in the Arctic is more than three times the global average increase,". (P2, L2-3)

**L3: 'decline in Arctic sea ice' be more specific, what is the parameter? Volume? Extent? What is the season you are referring to? Annual? summertime?**

Reply> The original expression "resulting in a rapid decline of Arctic sea ice and extreme cold events," has been modified to "resulting in a rapid decline of Arctic sea ice extent in all months, a decrease in extreme cold events,". (P2, L3)

**L5: 'SLCFs' to SCLF**

Reply> We think "SLCFs" is more appropriate. (P2, L6)

**L6: 'has' to have**

Reply> The word "has" has been changed to "have". (P2, L6)

**L6: Can you not refer to the more updated AMAP reports?**

Reply> Yes, the "AMAP Assessment 2021: Impacts of Short-lived Climate Forcers on Arctic Climate, Air Quality, and Human Health" has been added to the reference list as "AMAP, 2021b". (P2, L7).

*L6: Arctic aerosol chemical*

Reply> The phrase "Aerosol chemical composition" has been changed to "Arctic aerosol chemical composition". (P2, L7)

*L7: American spelling: 'sulphate' to sulfate*

Reply> The word "sulphate" has been changed to "sulfate". (P2, L8)

*L7: 'sea-salt' to sea-spray aerosol*

Reply> We think "sea-salt" is more appropriate because the subject of the sentence is "Arctic aerosol chemical composition". (P2, L8)

*L9: causes direct and/or semi-direct or can cause direct and semi-direct climate forcing*

Reply> The expression "direct/semi-direct" has been modified to "direct and/or semi-direct". (P2, L9-10)

*L9: it is not just BC acting as cloud condensation nuclei (CCN) – it makes it seem like this by the way you single BC out.*

Reply> The original sentence "Particularly, BC aerosols in the Arctic atmosphere can absorb solar radiation directly which causes direct/semi-direct climate forcing and work as cloud condensation nuclei (CCN) which causes indirect climate forcing (McFarquhar et al., 2011)." has been modified to "Particularly, BC aerosols in the Arctic atmosphere can absorb solar radiation directly which causes direct and/or semi-direct climate forcing (AMAP, 2011). Besides, BC aerosols can also act as cloud condensation nuclei (CCN) which causes indirect climate forcing (AMAP, 2011; McFarquhar et al., 2011).". (P2, L8-11)

*L11: change 'radiation' to radiation budget and 'besides' to leading to an acceleration in the melting of snow and ice*

Reply> The original sentence "When deposited onto snow/ice surface, BC can also affect the radiation due to reduction of the surface albedo; besides, it can accelerate the snow/ice melting due to its light absorption ability." has been modified to "When deposited onto snow/ice surface, BC can also affect the radiation budget due to reduction of the surface albedo, leading to an acceleration in the melting of snow and ice (AMAP, 2011).". (P2, L11-13)

*L15: remove the word 'warranted'*

Reply> The word "warranted" has been changed to "critical". (P2, L16).

*L16: 'in observatories' to at ground-based Arctic observatories*

Reply> The phrase "in observatories" has been modified to "at ground-based Arctic observatories". (P2, L17)

*17: 'Barrow/Utqiagvik' choose one or write formerly called Barrow for example*

Reply> The expression "Barrow/Utqiaġvik" has been modified to "Utqiaġvik". (P2, L18)

*L18. 'whereas those data' to These mong-term data sets provided…*

Reply> The expression "those data" has been modified to "these long-term datasets". (P2, L19)

*L22: 'Airborne observations, especially aircraft-based ones, earth-surface conditions' – not completely understood what is meant by this. Please rephrase.*

Reply> The original sentence "Airborne observations, especially aircraft-based ones, are less constrained from earth-surface conditions and allow the vertical profiles of BC in different seasons to be evaluated (e.g., Schulz et al., 2019; Ohata et al., 2021a; Jurányi et al., 2023)." has been modified to "Airborne observations have illustrated the vertical distributions of BC above the Arctic Ocean surface (e.g., Schulz et al., 2019; Ohata et al., 2021a; Jurányi et al., 2023).". (P2, L22-24)

*L22: Why 'different seasons' makes it soon like the reason for airborne observations are to expand on the number of seasons in which measurements can be performed*

Reply> The original sentence "Airborne observations, especially aircraft-based ones, are less constrained from earth-surface conditions and allow the vertical profiles of BC in different seasons to be evaluated (e.g., Schulz et al., 2019; Ohata et al., 2021a; Jurányi et al., 2023)." has been modified to "Airborne observations have illustrated the vertical distributions of BC above the Arctic Ocean surface (e.g., Schulz et al., 2019; Ohata et al., 2021a; Jurányi et al., 2023).". (P2, L22-24)

*L23: 'Shipborne observations allow for in situ measurements in the remote Arctic Ocean especially in summer and autumn when the Arctic sea ice is at the minimum' – Shipborne observations allow for measurements in Arctic Ocean at all times of the year, it is just that in summer and autumn these measurements are easier to preform as the Arctic Ocean is more accessible.*

Reply> The original sentence "Shipborne observations allow for in situ measurements in the remote Arctic Ocean especially in summer and autumn when the Arctic sea ice is at the minimum" has been modified to "Meanwhile, shipborne observations have facilitated in situ measurements in the remote

Arctic Ocean, especially in summer and autumn when the Arctic sea ice is at the minimum, making access to the Arctic Ocean easier". (P2, L24-26)

*L28: 'over' to 'in'*

Reply> The word "over" has been changed to "in". (P2, L29)

*L29: remove the word 'preliminary'*

Reply> The original expression "These shipborne studies have provided preliminary results of BC mass concentrations for" has been modified to "These shipborne studies have provided BC mass concentration results used for". (P2, L30)

*L30: Arctic Ocean?*

Reply> The phrase "Arctic Seas" has been modified to "Arctic Ocean". (P2, L31)

*L30-33: Break up sentence too long*

Reply> The original sentence "They also revealed important characteristics of the spatial distribution of BC in the Arctic Ocean and its marginal seas such as the BC concentration decreases with the growing distance from the source region (Xie et al., 2007; Sakerin et al., 2015, 2021)." has been modified to "They also revealed important characteristics of the spatial distribution of BC in the Arctic Ocean, demonstrating that BC concentration diminishes in the northern direction and decreases as distance from the continent increases (Xie et al., 2007; Sakerin et al., 2015, 2021).". (P2, L31-P3, L1)

*L32: 'decreases with the growing distance from the source region' – remove 'the growing' and replace with 'increasing'*

Reply> The expression "with the growing distance from the source region" has been modified to "as distance from the continent increases". (P2, L33)

*Page3:*

*L1: Mention the Arctic Haze phenomenon or refer to*

Reply> The original expression "with high values in winter and low values in summer." has been modified to "with high values in winter and spring – the Arctic Haze season (Barrie, 1986), and low values in summer and early autumn.". (P3, L2-3)

*L1: Relatively 'large' what BC mass concentrations*

Reply> By "relatively large" we mean that the seasonal change during MOZAiC observation is larger than that at ground-based observatories. We are sorry for the unclear expression. For clarity, the phrase "relatively large" has been modified to "the changes are larger". (P3, L2)

*L2: 'those' to 'these studies were limited'*

Reply> The expression "those studies limited" has been modified to "these studies were limited". (P3, L4)

*L4-5: the parts of the seas which are close to land or are you simply mentioning that these seas are close to land (if so not really necessary)*

Reply> The phrase "close to land" has been deleted. (P3, L6)

*L9-13: mention the importance of preforming measurements in regions with sparse coverage*

Reply> This information is included in the revised manuscript by adding ", where data coverage is sparse," after "especially in the western central Arctic Ocean and East Siberian Sea". (P3, L14)

*L16: Remove 'whereas models have been improving in the past two decades'*

Reply> "Whereas models have been improving in the past two decades" have been deleted. (P3, L18)

*L18: 'The main obstacles include poor understanding in' replace in with of*

Reply> The word "in" has been replaced with "of". (P3, L19)

*L24: 'in the context of climate change, the likelihood of extreme Arctic fire weather will increase.' – what do you mean by 'Arctic fire weather' this term is unknown to me.*

Reply> The expression "extreme Arctic fire weather" has been modified to "extreme fire weather in the Arctic". (P3, L26)

*L25: the impact on BC emissions*

Reply> We think that "the impact of BC emissions … on …" is appropriate, so we have retained it without alternations. (P3, L26-27)

*L27: 'is' replace with 'are' as you mention 'studies'*

Reply> The word "is" has been changed to "are". (P3, L29)

*L29: 'BC monitors based on light absorption theory' rephrase'*

Reply> This information has been removed from the revised manuscript. (P3, L30)

*L1-4: No aims of the study are listed but the results are described*

Reply> The first sentence of the paragraph "In this study, BC monitors based on light absorption theory were operated during the 2016–2020 summer and autumn expeditions from the North Pacific Ocean to the Arctic Ocean, encompassing the western Arctic Ocean and part of the east Siberian Sea, and back to the North Pacific Ocean to measure the BC mass concentration ($m_{BC}$)." has been modified to

"In this study, to enhance comprehension of the distribution and sources of BC in the Arctic, the mass concentration of BC ($m_{BC}$) was monitored across five round-trip expeditions conducted between the North Pacific Ocean and the Arctic Ocean during the summer and early autumn of 2016–2020." (P3, L30-32)

*Shipborne observations:*

*L8: Makes it sound like this is the only thing that was done to clean the data from the ship exhaust, however, later on you mention other steps.*

Reply> The original expression "The air intake was set at the handrail of the front upper deck to avoid ship exhaust pollution." has been modified to "The air intake was set at the handrail of the front upper deck to prevent contamination from ship exhaust pollution. Furthermore, detailed information regarding data filtering techniques to mitigate the impact of ship exhaust will be provided later.". (P4, L9-11)

*L11: Would be nice to mention clear how many expeditions were carried out in total even though it is clear from seeing the results and assuming that one cruise was carried out each year.*

Reply> This information has been added to the last paragraph of the Introduction section. (P3, L31)

*L13: Mention the concept of the mass absorption cross-section (MAC) here as this is an important concept*

Reply> The MAC values used for the two instruments have been included in the revised manuscript as follows:

For COSMOS, "The default mass absorption cross section (MAC) of 10 $m^2$ $g^{-1}$ was applied for the derivation of $m_{BC}$." (P4, L31-32)

For AE22, "The default manufacturer-provided MAC value of 16.6 $m^2$ $g^{-1}$ was applied." (P5, L11)

*L14: Why is it referred to as 'BC' when what you are measuring is eBC?*

Reply> We apologize for the misleading explanation in the preprint. In the revised manuscript, a detailed explanation regarding this is added as follows:

"Whereas both instruments use light absorption methods, COSMOS was equipped with a 400 °C heated inlet line. This feature effectively eliminated interference from volatile non-refractory aerosol chemical species internally mixed with BC, ensuring a high accuracy of $m_{BC}$ measurement. This aspect has been critically assessed in previous studies (Ohata et al., 2019; Sinha et al., 2017). Consequently, COSMOS measurements differ from traditional light absorption methods, where the mass concentration of BC is referred to as equivalent BC (eBC, Petzold et al., 2013). Therefore, instead of using eBC, the term BC can be used for COSMOS data in a general sense (Ohata et al., 2019). Henceforth, when comparing data from the two different instruments, we will use $m_{eBC}$ to represent the BC mass concentration measured with the Aethalometer during the 2017, 2018, and 2020 cruises, and $m_{BC}$ (COSMOS) to represent the BC mass concentration measured with COSMOS during the 2016–2019 cruises. Otherwise, BC mass concentration is denoted as $m_{BC}$ for simplicity." (P4, L15-25)

In addition, a comparison between the two instruments, combined with previous studies led us to decide to use Aethalometer data mainly as a reference. This point is included in the revised manuscript as follows: "Comparison between $m_{eBC}$ and $m_{BC}$ (COSMOS) for cruises in 2017 and 2018, when both data are available, shows that the two data are in high consistency (Pearson correlation coefficient $R>0.96$) and that $m_{eBC}$ was 1.3–2.5 times $m_{BC}$ (COSMOS) (Fig. S2). Previous studies also show that the default parameter settings of the Aethalometer, as mentioned above, may cause the obtained BC mass concentrations to be 1–3 times the mass measured by SP2, depending on the sources and mixing states of the BC aerosols (Wang et al., 2014; Sharma et al., 2017; Laing et al., 2020). Due to the above reasons, the AE22 data in this study are mainly used as a reference." (P5, L12-19)

*L18: 'data integration time' what is this term and what does it refer to? Not completely clear.*

Reply> The phrase "data integration time" has been changed to "measurement interval". (P4, L30)

*L23: Why was the wavelength 880nm chosen? Why not the wavelength most similar to 565nm, can you justify your reasoning?*

Reply> The 2nd AE22 channel uses 370 nm, which is found to respond with great sensitivity to aromatic organic species, and thus is not suitable to compare directly with the COCMOS 565 nm BC data.

*L26: Is this MAC value site-specific i.e. for the Arctic? Is there a more relevant MAC value?*

Reply> The manufacturer-provided MAC value for AE22 at 880 nm is 16.6 $m^2$ $g^{-1}$, and this value has been incorporated into the revised manuscript. We opted for the use of this single value over applying a correction method (e.g., Backmann et al., 2017) due to the wide range of latitudes covered in the measurements conducted in this study.

*L28: default parameter settings such as …*

Reply> The default parameter settings mainly refer to the default manufacturer-provided MAC value of 16.6 m$^2$ g$^{-1}$ of AE22. The original expression "It is noted that the default parameter settings of the aethalometer may cause the obtained BC mass concentrations to be twice the actual values (Laing et al., 2020; Asmi et al., 2021)." has been modified to

"Previous studies also show that the default parameter settings of the Aethalometer, as mentioned above, may cause the obtained BC mass concentrations to be 1–3 times the mass measured by SP2, depending on the sources and mixing states of the BC aerosols (Wang et al., 2014; Sharma et al., 2017; Laing et al., 2020).". (P5, L15-18)

*L28: Aethalometer*

Reply> The expression "aethalometer" has been modified to "Aethalometer". (P5, L16)

*Page 5:*

*L6: 'lump' replace with lamp*

Reply> The word "lump" has been replaced with "lamp". (P5, L30)

*L21-22: why impose this criterion of 40 minutes of valid data records. Does that mean that at least 40/5 = 8 5-min values or 40 1-minute values are needed.*

Reply> We use standards that require 40 minutes of valid data recording to ensure that each hour's data accurately represents that corresponding hour. Yes, at least 8 5-min values or 40 1-min values are required to calculate hourly values. (P6, L15-16)

*Model simulations:*

*L31: When was the data accessed? Why were small fires not included?*

Reply> The inclusion of small fires in the model was unintentionally omitted in the manuscript. The original expression "The Global Fire Emissions Database (GFED v4.1)" has been revised to "The Global Fire Emissions Database with small fires (GFED v4.1s)." (**P6, L22-23**) We apologize for any confusion.

*Page 6:*

*L5: Expand on the criteria you used for the HYSPLIT runs. Defend and state the number of hours you ran the model for? What was the initialised height? What was the temporal resolution for the runs? How did you track the movement of the ship using the model?*

Reply> We didn't use the model to track the movement of the ship. Other responses regarding this comment have been addressed in our response to the reviewer's first major comment.

*Line 18-20: unclear what is achieved by the groupings and they are not used so much afterwards:*

*South of 52°N: (in the North Pacific Ocean),*

*North of 72° N: (mainly in the Canada Basin and the east part of the East Siberian Sea, which are noted as western central Arctic Ocean in the following sections of this study),*

*between 52 and 72° N: (mainly in the Bering, Chukchi, and Beaufort Seas).*

Reply> The groupings are used to characterize the spatial-temporal variations of $m_{BC}$ and the contribution of biomass burning to the observed BC along the cruise route. The main findings are a decrease in $m_{BC}$ with increasing latitude (Paragraph 2, Sect. 4.1) and an increase in the contribution of biomass burning to the observed BC as latitude increases (Paragraph 1, Sect. 4.4).

Please note that the grouping mentioned here does not comply with the latitudinal constraints (i.e., north of 65° N) used to select high BC episodes in the Arctic. This information has been added to the second paragraph of Sect. 4.1 as follows: "Note that the grouping mentioned here does not comply with the latitudinal constraints (i.e., north of 65° N) used to select high BC episodes in Sect. 4.4.". (P7, L25-27)

*Page 7-8:*

*L4: remove 'pre-existing'*

Reply> The phrase "pre-existing" has been removed. (P8, L11)

*L34-page8, L1: 'necessity to further study the spatial-temporal variations of BC in the Arctic Ocean' move to discussion*

Reply> The sentence "This also indicates the necessity to further study the spatial-temporal variations of BC in the Arctic Ocean." has been removed from the paragraph.

*L3: 'shouldn't' to should not*

Reply> The expression "shouldn't" has been modified to "should not". (P9, L11)

*L7: 'Barrow' to 'Utqiagvik'*

Reply> The word "Barrow" has been changed to "Utqiaġvik". (P10, L3)

*L10: Is it possible to know whether BB reach further north or that the ratio increases because the anthropogenic influence decreases at the latitude increases?*

Reply> Yes, it is possible by comparing the spatial distribution of anthropogenic BC and biomass burning BC based on the model result as illustrated in Sect. 4.4 in this study. In the revised manuscript, this information is more clearly represented by adding references to previous studies: "This aligns with previous model studies indicating that during summer, the transport efficiency of low latitude anthropogenic BC to the Arctic was low, and biomass burning BC contributed more than 63 % to the surface BC in the Arctic (Ikeda et al., 2017; Zhu et al., 2020).". (P17, L25-28)

*Page 9:*

*Figure 2: The smaller values are not very clear. Seems as though values are cut-off – represent or detail in the caption the values that you cut-off. The font size for the y-axis can be increased. Remove 'Year/month/day' not needed. You could separate the plot based on latitudes and focus on the events which are all above 65 degrees north. Not 100% clear what time resolution is presented here.*

Reply> Figure 2 has been revised to include individual data in panel (a). A statement "All data presented here is at 1 h time resolution and the data influenced by ship exhaust has been removed." has been added to the caption. The label for the bottom axis has been removed. The full-scale panel (a) and a zoomed-in view of panel (a) with the y-axis maximum set to 80 ng m$^{-3}$ are now shown in Fig. S3.

*4.2 Background BC concentration in the western central Arctic Ocean*

*L9: remove 'pristine region' not scientific or perhaps rephrase to distant from sources of anthropogenic pollutants*

Reply> We apologize for the misleading information in the preprint. The original expression "Ideally, the Arctic Ocean is one of the pristine regions in the world and background concentrations of primary air pollutants such as black carbon there should be close to zero. However, zero background black carbon may not be true to the Arctic marine boundary layer atmosphere due to the influence of" has been modified to

"While finding a situation entirely identical to the preindustrial atmosphere is challenging due to the pervasive influence of anthropogenic activities on even natural events like wildfires (McCarty et al., 2021), examining periods in the Arctic Ocean unaffected by regional transport could offer insights into the preindustrial atmospheric situations. This assumes that the impact of natural terrestrial activities, such as wildfires, on BC in the preindustrial Arctic Ocean atmosphere was likely negligible, recognizing the inherent uncertainties in making such historical assessments." (P15, L9-14)

*L10-13: 'should be close to zero' why? What does 'close to zero' mean? Use the literature to suggest typical values as opposed to some arbitrary 'close to zero' statement which is quite meaningless. Who claims that the marine Arctic boundary layer should have a 'zero background'? BC has been transported up to the Arctic prior to pre-industrial times (McConnell et al., 2007). McConnell et al., (2007) present historical BC from Greenland ice-cores showing non-zero values.*

Reply> We apologize for the misleading information in the preprint. The original expression "Ideally, the Arctic Ocean is one of the pristine regions in the world and background concentrations of primary air pollutants such as black carbon there should be close to zero. However, zero background black carbon may not be true to the Arctic marine boundary layer atmosphere due to the influence of" has been modified to

"While finding a situation entirely identical to the preindustrial atmosphere is challenging due to the pervasive influence of anthropogenic activities on even natural events like wildfires (McCarty et al., 2021), examining periods in the Arctic Ocean unaffected by regional transport could offer insights into the preindustrial atmospheric situations. This assumes that the impact of natural terrestrial activities, such as wildfires, on BC in the preindustrial Arctic Ocean atmosphere was likely negligible, recognizing the inherent uncertainties in making such historical assessments." (P15, L9-14)

*L13: 'anthropogenic productive' replace with 'industry'*

Reply> The phrase "anthropogenic productive" has been replaced by "industry". (P15, L16)

*L14: lower latitude*

Reply> The phrase "low latitude" has been modified to "lower latitude". (P15, L16)

*L14: 'export' replace with transport*

Reply> The word "export" has been replaced by "be transported". (P15, L17)

*L15: how do commercial fisheries significantly impact Arctic BC?*

Reply> The phrase "commercial fisheries and" has been removed. (P15, L19)

*L17: rephrase the sentence 'coastal region along the Arctic climate warming' doesn't make sense*

Reply> The original expression "along the Arctic climate warming" has been modified to "driven by the warming Arctic climate". (P15, L19)

*L18: which allows for the transport…*

Reply> The expression "which allows transport…" has been modified to "which allows for the transport…". (P15, L21)

*L19: Separate sentence/start a new sentence: 'and the stable atmospheric conditions'*

Reply> The original sentence "In winter and early spring, the buildup of anthropogenic pollutions, due to the expansion of the polar dome, which allows transport of anthropogenic pollutants from continental regions further south, and the stable atmospheric conditions, can lead to monthly mean $m_{BC}$ of as high as more than 100 ng m$^{-3}$ (e.g., Boyer et al., 2023)." has been modified to "In winter and early spring, the buildup of terrestrial anthropogenic and natural pollutants occurs due to the expansion of the polar dome, which allows for the transport of pollutants from continental regions further south. This buildup, combined with stable atmospheric conditions, can result in monthly mean $m_{BC}$ levels exceeding 100 ng m$^{-3}$ (e.g., Boyer et al., 2023).". (P15, L20-23)

*L20: replace with higher*

Reply> The expression "of as high as more than" has been changed to "levels exceeding". (P15, L23)

*L21: surface layer atmosphere*

Reply> The original expression "Arctic Ocean surface atmosphere" has been modified to "Arctic Ocean surface layer atmosphere". (P15, L25)

*L24-25: Why are summer and early autumn months most suitable to evaluate background levels? What are you trying to achieve? By background do you mean pre-industrial – the background will have a seasonality as described by the Arctic Haze phenomenon.*

Reply> We appreciate your precious questions and comments. We apologize for the misleading in our previous definition of the background periods.

Yes, the background in this context refers to the preindustrial Arctic Ocean conditions, which could have seasonality due to changes in atmospheric transport through the year. In the revised manuscript, we modify the definition of the background period to be a period in the Arctic Ocean that is not affected by regional transport, which is more likely to occur during summer and early autumn. Please see Sect. 4.2 for details.

*L26: The definition of 'background periods' needs to be properly explained it is not clear why the criteria you mentioned have been applied. What is it you want to achieve with this notion of background (perhaps unaffected by anthropogenic emissions?). Background Arctic Ocean conditions perhaps is better wording. The definition relies on values below the detection limit 'below the lowest*

*detection limit' there is it valid? You could additional work out how often the measurements exceed these background values.*

Reply> We apologize for the misleading in our previous definition of background period. The background we intended to define was the background Arctic Ocean conditions which was not influenced by regional transport of air pollutants from terrestrial regions. This situation was more likely to occur in summer than winter months and may represent the preindustrial summer conditions in the Arctic Ocean. Therefore, we have revised the definition of Arctic Ocean background periods from "The background periods in the western central Arctic Ocean ($>72°$ N) were defined according to the $m_{BC}$ measured by COSMOS and the 5 day HYSPLIT back trajectories as follows: the 1-min $m_{BC}$ was below the lowest detection limit of 50 ng m$^{-3}$ for continually 2 hours or longer, the 1-h $m_{BC}$ was above the lowest detection limit of 1 ng m$^{-3}$, and the air masses were from the Arctic Ocean." to

"The background periods in the western central Arctic Ocean ($>72°$ N) were determined according to the following criteria: first, for each hour with effective BC data, all three 5-day HYSPLIT back trajectories initiated at starting heights of 10, 500, and 1000 m originated from the Arctic Ocean. Additionally, all 1-min $m_{BC}$ or 5-min $m_{eBC}$ data within that hour were not removed due to ship exhaust according to data screening criteria described in Sect. 2. The second criterion is to ensure the accuracy of the selected data.". (P15, L30-34)

*L28: remove 'continually'*

Reply> The entire sentence was deleted from the revised manuscript.

*Page 12:*

*L2: replace 'imported' with long-range transport*

Reply> The comparison between estimated background $m_{BC}$ and central Arctic Ocean $m_{BC}$ has been deleted from the revised manuscript.

*L3-4: 'might be twice the actual values because the default' this is repeated*

Reply> The comparison between estimated background $m_{BC}$ and central Arctic Ocean $m_{eBC}$ has been deleted from the revised manuscript.

*L7: There are decreasing trends in the Arctic and this has been well reported and hence is representative of this time period e.g. Hirdman et al., 2010, Sharma et al., 2004,*

Reply> Although previous studies reported a decreasing trend in winter eBC, the annual trend in summer eBC remains uncertain (e.g., Sharma et al., 2004; Schmale et al., 2022). Could the reviewer be more specific with the comment? Thank you so much.

*L9: Unsure how the background is calculated.*

Reply> We apologize that we haven't stated this point clearly. The original expression "The overall mean of background $m_{BC}$ calculated from $m_{BC}$ at 1 h time resolution for 53 hours was 3.3 ($\pm$1.5) ng m$^{-3}$ (Table 1)." has been modified to "The overall mean of background $m_{BC}$, calculated from COSMOS $m_{BC}$ at 1 h time resolution over 52 hours, was 7.5 ($\pm$8.5) ng m$^{-3}$." (P16, L10-11)

*L17: Why not R2?*

Reply> We present the Pearson correlation coefficient, denoted as $R$. Hence, we will consistently use $R$ instead of showing $R^2$.

*Page 13, 4.4 Sources of High BC episodes:*

*In 4.4. Sources of High BC episodes, the phrasing is not consistent, these episodes are referred to as 'high BC episodes', 'elevated BC mass concentration periods' and 'high BC mass concentration episodes.*

*Winiger et al., 2016 might be a good study to read and reference.*

Reply> We appreciate the paper the reviewer recommends.

In the revised manuscript, these episodes are uniformly referred to as high BC episodes. Accordingly, the following revisions are made in the revised manuscript:

"elevated-BC episodes" → "high BC episodes". (P1, L24)

"high BC mass concentration episodes" → "high BC episodes". (P17, L33)

"elevated BC mass concentration episodes" → "high BC episodes". (P29, L22)

*L12 – 15: Why are the elevated values defined as such what is the reasoning? i.e. why the criteria '1-h $m_{BC}$ was continually greater than 10ngm$^{-3}$ for 18 h or longer and the mean of valid 1-h $m_{BC}$ during the defined periods was greater than 20 ngm$^{-3}$. Why 18 h? Why 10 ngm$^{-3}$, why 20 ngm$^{-3}$? What is the motivation?*

Reply> The criterion of 10 ng m$^{-3}$ represents three times the background $m_{BC}$ level determined in Sect. 4.2 of the preprint. Furthermore, considering that longer episodes and higher BC mass concentrations

would offer better representation, we refined the selection criteria to ultimately identify approximately 10 episodes. In the revised manuscript, we have retained this definition but modified the presentation.

The original expression regarding the definition of high BC episodes "To characterize the sources of the high concentrations of BC in the Arctic Ocean and the marginal seas, high BC episodes were defined as periods when the 1-h $m_{BC}$ was continually greater than 10 ng m$^{-3}$ for 18 h or longer and the mean of valid 1-h $m_{BC}$ during the defined periods was greater than 20 ng m$^{-3}$. In total, 10 high BC mass concentration episodes were identified (Figs. 1, 3, S1, and S3, Table 2)." has been modified to "To characterize the sources of the high concentrations of BC in the Arctic Ocean and the marginal seas (north of 65° N), we identified periods when the 1-h $m_{BC}$ exceeded 10 ng m$^{-3}$. From these periods, we further selected those lasting 18 h or longer and the mean of valid 1-h $m_{BC}$ during the selected period was not less than 20 ng m$^{-3}$. This process allowed us to identify and refine 10 high BC episodes (Figs. 1, 3, S1, and S6, Table 3).". (P17, L29-P18, L1)

*L21-22: 'Note that in addition to these 10 episodes, high BC was also observed in the Arctic at other times, such as on 14-18 August 2020' – is this sentence needed?*

Reply> This sentence has been removed from the revised manuscript.

*L25: support what you mean by 'well reproduced'?*

Reply> What we wanted to say here is that both observed and model data showed peaks during these episodes. Therefore, the original expression "Furthermore, the temporal and spatial variations of E3, E8, and E10 were well reproduced by GEOS-Chem model (Fig. 3)." has been complemented as "Additionally, the temporal and spatial variations of E3, E8, and E10 were well reproduced by GEOS-Chem model, showing nearly simultaneous peaks in observed and model data during these episodes (Fig. 3).". (P18, L14-16)

*Page 14:*

*L6: replaced 'occurred' with was measured*

Reply> The expression "occurred" has been modified to "was measured". (P19, L6; P20, L11; and P26, L12)

*L11: Remove 'Contour plots'*

Reply> The expression "Contour plots in Fig." has been changed to "Figure". (P19, L11; and P21, L7)

*Page 16:*

*L3: rephrase '… burning BC and surface winds before to after Episode 8'*

Reply> The expression "… burning BC and surface winds before to after Episode 8" has been modified to "… burning BC and winds before, during, and after Episode 8". (P21, L4)

*5 Summary and conclusions:*

*Page 23:*

*L2 'higher than low latitude' – what does this mean?*

Reply> The original sentence "Relatively low $m_{BC}$ were observed at higher than low latitude regions." has been modified to "Relatively low levels of $m_{BC}$ were observed at higher latitude regions.". (P29, L5-6)

*L8-10: Repeated definition*

Reply> This part has been deleted from the revised manuscript.

*L24: Arctic BC mass*

Reply> The expression "observed Arctic BC" has been modified to "observed Arctic BC mass". (P29, L32)

---

## Author Response (AR2)

**Response to comments from reviewer 1**

We appreciate valuable comments on our manuscript by the reviewer. Our responses are listed below.

*Review of the manuscript titled 'Measurement report: Shipborne observations of black carbon aerosols in the western Arctic Ocean during summer and autumn 2016–2020: impact of boreal fires.'*

*The manuscript significantly improved after the first revision. I have a few minor comments as indicated below.*

*Minor comments:*

*Line 7: Several studies were conducted to understand the chemical composition of aerosols over the Arctic. The authors can use the literature here. I would recommend modifying this sentence.*

Reply> The original expression "Arctic aerosol chemical composition may include black carbon (BC), sulfate ($SO_4$), nitrate ($NO_3$), organics, sea-salt, and mineral dust." has been modified to "Arctic aerosol chemical composition may include black carbon (BC), sulfate ($SO_4$), nitrate ($NO_3$), organics, sea-salt, and mineral dust (Sakerin et al., 2015; AMAP, 2021b; Schmale et al., 2022).". (P2, L7-9)

*Page 5, Line 5: What authors would like to convey here by discussing SP2 as a reference instrument in the previous studies? Did the authors use any of these datasets in this study?*

Reply> By discussing SP2 as a reference instrument, we intend to rationalize the comparisons between COSMOS and SP2 (P5, L2-5) and between Aethalometer and SP2 (P5, L19-22). We didn't use any SP2 data in this study.

The original expression "The SP2 is used as a reference instrument in previous studies (e.g., Ohata et al., 2019; Sinha et al., 2017)." has been modified to "The SP2 was often used as a reference instrument in previous studies (e.g., Ohata et al., 2019; Sinha et al., 2017).". (P5, L5-6)

*Page 5, Line 12: How did the authors estimate the lower detection and error limits? I would suggest adding the details here. Also, explain the noise level of the Aethalometer measurements in this study and how the authors treated them while processing the data. I would also like to mention here the multiple scattering effects in the Aethalometer datasets.*

*The authors did not mention it in the manuscript and the applied correction procedures if they had done it.*

Reply> To confirm the performance of the Aethalometer, the manufacturer conducted a 24-hour particle-free zero air testing. The results demonstrated that the instrument met the 24-hour zero air mean detection limit of 20 ng/m³ and a 5-minute zero air standard deviation limit of ±30 ng/m³, as stated in the instrument manual (https://www.psi.ch/sites/default/files/import/catcos/ProjectDetailCatcosOperationsEN/Aethalometer_book_2005.07.02.pdf). To minimize noise levels, we averaged the 5-minute data to 1-hour intervals for further analysis. The multiple scattering effects in the Aethalometer datasets were not corrected, which could be one of the main reasons for the large uncertainty compared to COSMOS and SP2, as discussed in the second half of the paragraph.

The previous expression "The data integration time was set to 5 min, which was then averaged to 1 h for further analysis. The default manufacturer-provided MAC value of 16.6 $m^2\ g^{-1}$ was applied. The lower detection limit of the Aethalometer at 1 h time resolution is 20 ng $m^{-3}$ with the error limit of ±30 ng $m^{-3}$." has been modified to "The data integration time was set to 5 min. For further analysis, hourly averages were used to minimize noise levels under clean atmospheric conditions. The default manufacturer-provided MAC value of 16.6 $m^2\ g^{-1}$ was applied for all analyses since the study area covers a wide range of latitudes. The manufacturer's particle-free zero air testing meets a 24-h mean detection limit of 20 ng $m^{-3}$ and a 5-min standard deviation limit of ±30 ng $m^{-3}$.". (P5, L12-16)

*Page 6, Line 5: Is there any specific reason for screening the data specifically for 3 ms-1.*
Reply> We determined the data screening criteria by experimenting with various criteria as well as referring to previous studies, such as Taketani et al. (2016). Other criteria we tested include requiring the 1-min wind direction and speed relative to the ship's course to be within ±70° of the bow and >2 m $s^{-1}$, respectively. Ultimately, we selected wind direction within ±60° of the bow and a wind speed >3 m $s^{-1}$. This choice effectively screened out data associated with simultaneous significant decreases in $O_3$ and increases in BC observed in the open North Pacific Ocean (approximately north of 42° N) and the Arctic Ocean.

*Page 6, Line 20: The authors used the Global Fire Emissions Database with small fires (GFED v4.1s) with 0.25° × 0.25° of spatial resolution in this study. This data is available only*

*up to 2016. But, this study focused on 2016- 2020. How does it affect the results of this study?*

Reply> GFED v4.1s data is available from 1997 through the present. For details, the reviewer may refer to these websites: https://www.globalfiredata.org/data.html and https://www.geo.vu.nl/~gwerf/GFED/GFED4/Readme.pdf

*Page 7, Line 5: Explain the reason behind estimating the airmass trajectories above 10 m above ground level in this study. I would expect the inlet position of the ship to be higher than this height. Also, I would recommend the authors use the satellite fire count data sets and the trajectories as separate figures in the supplementary details for each cruise period.*

Reply> We agree that the inlet position of the ship could be higher than 10 m above the model ground level. In this study, we only used the trajectories started at 500 m above the model ground level for BC source interpretation. And we used trajectories starting at 10, 500, and 1000 m above the model ground level for background period identification. This information has been added to the revised manuscript as shown below: "Note that for the source interpretation of the observed BC, only back trajectories starting at 500 m above model ground level were employed. Trajectories starting at 10, 500, and 1000 m above the model ground level were used for background period identification.". (P7, L11-13)

We have presented biomass burning BC emission maps and back trajectories for the three feature episodes as Figs. S9, S11, and S14. We have decided not to include satellite fire count data and back trajectories for all cruise periods in the supplemental file, as we believe it would not significantly contribute to the discussion points in the manuscript.

**Other minor changes:**

**(P6, L19-20)** The original expression "Within the hourly BC mass concentration data, 5–13 % of COSMOS data and 63–71 % of Aethalometer data fall below their respective detection limits." has been modified to "Within the hourly BC mass concentration data, 5–13 % of COSMOS data fall below its detection limit.".

**(P11, L3)** "mean(± 1 standard deviation)" has been modified to "mean(±standard deviation)".

**(P15, L4-5)** "The $m_{BC}$ presented here is at 1 h time resolution and the data influenced by ship exhaust has been removed." has been added to the caption of Fig. 3.

**(P26, L2)** "CO (b), $CO_2$ (c), and $CH_4$ (d) mixing ratios" has been modified to "$CH_4$ (b), $CO_2$ (c), and CO (d) mixing ratios".

**Text S1, Table S1, and Fig. S18** are changed due to changes in the background $O_3$ mixing ratios. Please refer to the track-change version for details.

---

## Author Response (AR3)

**A list of minor changes**

Data availability:

The original expression "...is available online (Deng et al., 2023; https://db.cger.nies.go.jp/MD/10.17595/202307XX.001.html.en; last access: 25 September 2023)."

has been changed to "is available online https://doi.org/10.17595/20240502.001 (Deng et al., 2024).".